# Mechanistic insight into the conserved allosteric regulation of periplasmic proteolysis by the signaling molecule cyclic-di-GMP

Debashree Chatterjee[1†‡], Richard B Cooley[1†], Chelsea D Boyd[2†§], Ryan A Mehl[3], George A O'Toole[2]*, Holger Sondermann[1]*

[1]Department of Molecular Medicine, College of Veterinary Medicine, Cornell University, Ithaca, United States; [2]Department of Microbiology and Immunology, Geisel School of Medicine at Dartmouth, Hanover, United States; [3]Department of Biochemistry and Biophysics, Oregon State University, Corvallis, United States

**\*For correspondence:**
georgeo@dartmouth.edu (GAO);
hs293@cornell.edu (HS)

[†]These authors contributed equally to this work

**Present address:** [‡]Molecular Biology Program, Sloan-Kettering Institute, New York, United States; [§]Department of Microbiology and Immunology, School of Medicine, University of North Carolina, Chapel Hill, United States

**Competing interests:** The authors declare that no competing interests exist.

**Reviewing editor**: Jon Clardy, Harvard Medical School, United States

**Abstract** Stable surface adhesion of cells is one of the early pivotal steps in bacterial biofilm formation, a prevalent adaptation strategy in response to changing environments. In *Pseudomonas fluorescens*, this process is regulated by the Lap system and the second messenger cyclic-di-GMP. High cytoplasmic levels of cyclic-di-GMP activate the transmembrane receptor LapD that in turn recruits the periplasmic protease LapG, preventing it from cleaving a cell surface-bound adhesin, thereby promoting cell adhesion. In this study, we elucidate the molecular basis of LapG regulation by LapD and reveal a remarkably sensitive switching mechanism that is controlled by LapD's HAMP domain. LapD appears to act as a coincidence detector, whereby a weak interaction of LapG with LapD transmits a transient outside-in signal that is reinforced only when cyclic-di-GMP levels increase. Given the conservation of key elements of this receptor system in many bacterial species, the results are broadly relevant for cyclic-di-GMP- and HAMP domain-regulated transmembrane signaling.

## Introduction

Biofilms are complex agglomerations of sessile, microbial cells surrounded in a self-secreted extracellular matrix that is composed primarily of exopolysaccharides, proteins, and nucleic acids (*Hall-Stoodley et al., 2004*). These communities are prevalent in natural as well as industrial and hospital settings and can form on a wide range of biotic and abiotic surfaces. Biofilm-forming pathogenic bacteria have been associated with numerous persistent and nosocomial infections in humans, such as infection of the ear or urinary tract or colonizing the lungs of patients suffering from cystic fibriosis (*Parsek and Singh, 2003*). Because bacteria in biofilms can withstand antibiotic treatment, many clinically relevant antibiotics are ineffective in the treatment of biofilm-related bacterial infections (*Mah and O'Toole, 2001*). Such antibiotic tolerance is a matter of concern especially in the context of the increasing number of multi-drug resistant strains and a rather slow rate of discovery of new antimicrobial agents (*Spellberg et al., 2004*). Hence, it is crucial to understand the molecular basis of biofilm formation, maintenance, and dispersal in order to identify novel targets that could potentially be used for disrupting these bacterial aggregates.

The decision to transition between a planktonic and a biofilm life-style is orchestrated by the near ubiquitous bacterial second messenger cyclic-di-GMP (c-di-GMP), a dinucleotide known to modulate many different aspects of bacterial physiology (*Ross et al., 1987*; *Hengge, 2009*). The dinucleotide is synthesized from two molecules of GTP by diguanylate cyclases (DGCs) containing a GGDEF domain, and hydrolyzed by phosphodiesterases (PDEs) with either an EAL or a HD-GYP domain (*Tal et al., 1998*;

**eLife digest** While bacteria often live as unicellular microorganisms, many bacteria are capable of sticking together on a surface and forming a multicellular structure called a biofilm. Bacterial biofilms occur frequently in nature; for example, on the roots of plants and submerged rocks. While these biofilms are generally innocuous, others pose significant health threats to humans, causing tooth decay, gum disease, and—when they occur on implanted devices such as prosthetic heart valves—potentially serious infections. When in biofilms, many bacteria are tolerant to antibiotics; therefore, working out how to disrupt these films is crucial for developing new treatments.

The microorganism *Pseudomonas fluorescens* is an example of a bacterium that can be found living in a complex biofilm. In response to certain environmental cues, free-swimming *P. fluorescens* cells adhere to a surface and produce a slime that encases them in a robust biofilm. The decision to shift between a free-swimming and a biofilm life-style is orchestrated by a signaling molecule found inside the bacteria called cyclic-di-GMP. In *P. fluorescens*, the availability of nutrients—in particular, phosphate—controls how much cyclic-di-GMP is produced inside the cell. If not enough phosphate is available, the level of cyclic-di-GMP falls and the biofilm disperses.

Cyclic-di-GMP affects the stability of the biofilm via a group of proteins called the Lap system. When levels of cyclic-di-GMP are high, cyclic-di-GMP binds to a protein called LapD, which can then in turn bind to an enzyme known as LapG. When bound to LapD, LapG is unable to break apart the molecules that help *P. fluorescens* cells bind to a surface, and so a biofilm can form. If cyclic-di-GMP levels drop, fewer LapD molecules can bind to cyclic-di-GMP. As cyclic-di-GMP-unbound LapD proteins interact poorly with LapG, this leaves some LapG molecules able to destabilize the attachments between the cells and the surface, which disperses the biofilm.

Here, Chatterjee et al. reveal the molecular mechanism by which LapD and LapG interact in *P. fluorescens*. When cyclic-di-GMP is bound to LapD, the shape of LapD changes to produce features that fit into the surface of LapG. It is this shape compatibility, more so than an increase in the number or quality of interactions between the chemical groups that make up the proteins, that enables LapD to bind to LapG. Chatterjee et al. also provide evidence that the LapD–LapG interaction can be disrupted, thereby raising the possibility that biofilm formation could be manipulated by targeting this system.

Given that systems similar to the *P. fluorescens* Lap system exist in numerous other bacterial species, including important pathogens, the findings of Chatterjee et al. could assist efforts to develop medicines and products that eradicate bacterial biofilms. LapD also shares many structural elements with a large number of other signaling proteins; therefore, these findings could also improve the understanding of how other cell signaling systems work.

*Simm et al., 2004*; *Ryan et al., 2006*; *Schirmer and Jenal, 2009*; *Krasteva et al., 2012*). Often these juxtaposing domains appear together in the same polypeptide chain, and in a particular subset, as part of enzymatically inactive multi-domain proteins providing them with the ability to sense and respond to changing levels of intracellular c-di-GMP. In *Pseudomonas fluorescens*, biofilm formation is dependent on such a signaling protein, referred to as LapD (*Hinsa and O'Toole, 2006*; *Newell et al., 2009*). This transmembrane receptor contains a cytoplasmic GGDEF–EAL domain module that lacks enzymatic activity, though the EAL domain has retained its capacity to bind c-di-GMP.

Cytosolic c-di-GMP levels in *P. fluorescens* are regulated in response to nutrient availability, specifically the amount of inorganic phosphate (*Figure 1A*; *Monds et al., 2007*). Limiting phosphate increases the expression of a PDE, which lowers intracellular c-di-GMP levels, ultimately resulting in the dispersion of *P. fluorescens* biofilms. Conversely, when environmental phosphate levels are high, multiple DGCs contribute to high intracellular c-di-GMP levels, resulting in stable cell attachment and subsequently biofilm formation (*Newell et al., 2011b*). LapD is a c-di-GMP receptor that translates changes in the concentration of this cytosolic second messenger into cell-surface events via an inside-out signaling mechanism that ultimately controls cell adhesion (*Newell et al., 2009*). Specifically, c-di-GMP-bound LapD engages the periplasmic protease LapG through its Per-Arnt-Sim (PAS)-like domain, preventing it from cleaving the cell surface-bound large adhesin LapA, a protein mediating cell adhesion to multiple substrates (*Navarro et al., 2011*; *Newell et al., 2011a*; *El-Kirat-Chatel et al., 2014*). In contrast, c-di-GMP-unbound LapD adopts an autoinhibited conformation and loses its affinity for

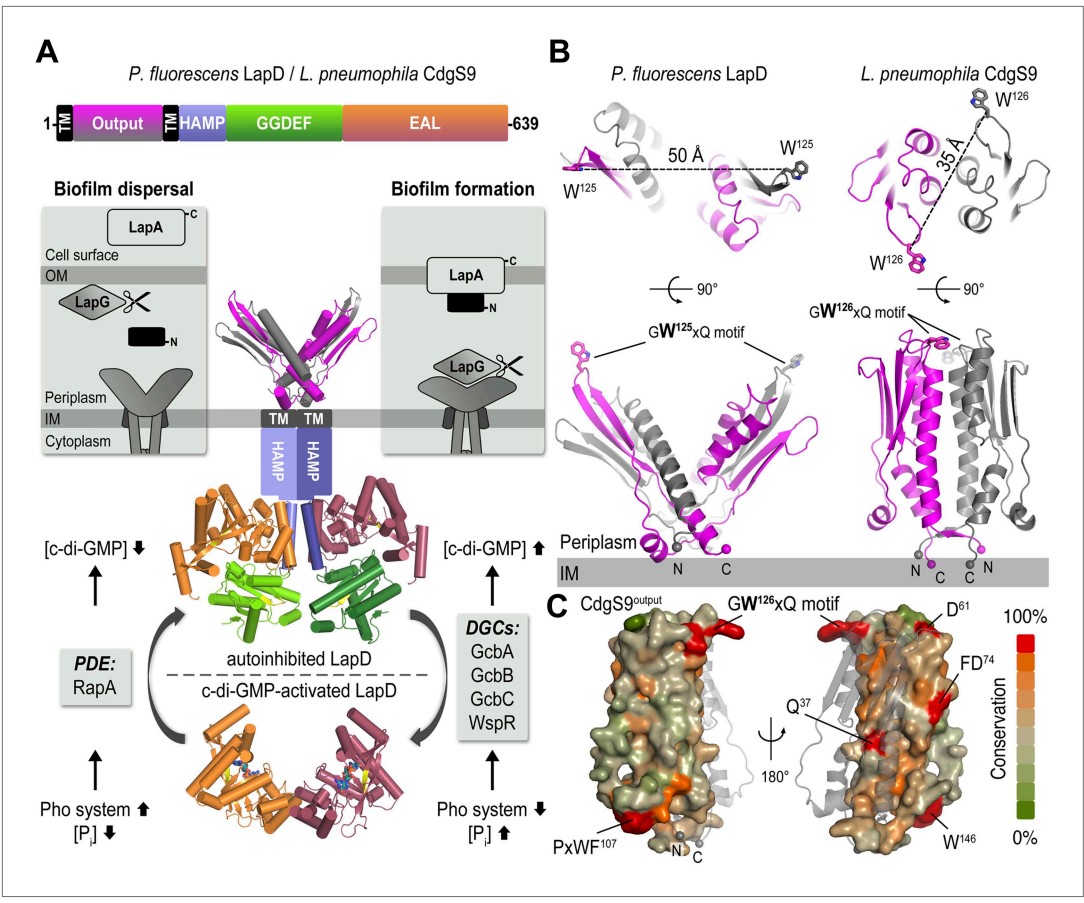

**Figure 1**. The LapD signaling system and structures of the LapD periplasmic domain. (**A**) Overview of LapD-mediated signaling. The primary structure and domain organization of the LapD receptor is shown (top panel). The cartoon (bottom panel) summarizes the current model of LapD-mediated regulation of biofilm formation in *P. fluorescens* by the differential recruitment of the periplasmic protease LapG in response to exogenous inorganic phosphate ($P_i$) availability (reviewed in ***Boyd and O'Toole, 2012***). Low bioavailability of $P_i$ induces the expression of phosphodiesterases (PDEs) via the Pho system, which results in low intra-cellular concentrations of c-di-GMP and LapD adopting an autoinhibited state. Auto-inhibited LapD cannot sequester the periplasmic protease LapG, allowing the LapG enzyme to cleave the cell-surface adhesin LapA, releasing it from the cell surface. Conversely, when environmental $P_i$ is present in sufficient concentration, a subset of diguanylate cyclases (DGCs) produce intra-cellular c-di-GMP to levels sufficient for binding to the EAL domain of LapD, thus activating this receptor. Activated LapD binds to and sequesters LapG in the periplasm, which promotes biofilm formation by preventing cleavage of LapA. (**B**) Crystal structures of the periplasmic output domain from *P. fluorescens* LapD and *L. pneumophila* CdgS9. Two orthogonal views of each output domain structure are shown in ribbon representation, with the two protomers colored pink and gray, respectively (*P. fluorescens* LapD[Output], PDB 3PJV; ***Navarro et al., 2011***). Distances between the tryptophan residues in the conserved GWxQ motif of each protomer are indicated. The relative position of the inner cell membrane (gray bar) and connection to the flanking transmembrane (TM) helices are indicated. (**C**) Surface conservation of the output domain. The sequences of 18 orthologs of LapD were aligned and the sequence conservation was mapped on to the solvent-accessible surface of CdgS9[Output]. The surface is colored according to the degree of conservation.

LapG. LapG in turn accesses and cleaves LapA's N-terminus, releasing the bulk of the adhesin from the cell surface and destabilizing cell attachment (***Boyd et al., 2012***; ***Chatterjee et al., 2012***; ***Boyd et al., 2014***). Thus, LapD acts as the central hub in *P. fluorescens* that converts environmental cues into the formation of or dispersal from biofilms.

Given that orthologs to LapD and LapG are found in a variety of bacteria, including several pathogens, understanding the molecular mechanism by which c-di-GMP binding to LapD's cytosolic EAL domain alters the ability of its periplasmic domain to bind LapG is of great interest. Previous work from

our groups has begun to establish some of these intrinsic regulatory principles (*Navarro et al., 2011*; *Newell et al., 2011a*). The PAS-like periplasmic domain of LapD that engages LapG is flanked by two putative transmembrane helices, the second of which connects to an intracellular, juxtamembrane HAMP domain followed by the aforementioned GGDEF-EAL domain module (*Figure 1A*). HAMP domains have been extensively studied in several other systems, as this domain is a crucial signal relay module in a large number of diverse proteins, not only in bacteria but also in archaea and eukaryotes (*Galperin et al., 2001*). These domains are known to be a homodimeric assembly adopting a parallel four-helix bundle (*Hulko et al., 2006*; *Ferris et al., 2011*), although the exact mechanism by which conformational changes are propagated through a signaling protein via HAMP domains remains a topic of great interest (*Swain and Falke, 2007*; *Zhou et al., 2009*; *Ames et al., 2014*). In LapD, the HAMP domain couples to the GGDEF-EAL domain module via a common extension of HAMP domains, the so-called signaling or S helix (*Anantharaman et al., 2006*). Our previous structure-function analyses revealed that the S helix docks onto the lateral surface of the EAL domain and positions the GGDEF domain above the c-di-GMP binding site at the EAL domain, preventing dinucleotide binding, and hence stabilizing the autoinhibited state. Conversely, c-di-GMP binding to the EAL domain is mutually exclusive with this autoinhibited conformation, disrupting inhibitory interactions, leading to an activation of LapD (*Navarro et al., 2011*). Through a poorly understood process, these conformational changes in the GGDEF-EAL domain module are sensed by the HAMP domain, which then propagates the signal to the periplasmic domain of LapD, ultimately resulting in a change in affinity for LapG (*Newell et al., 2009*; *Newell et al., 2011a*).

The elucidation of a complete functional circuit from nutritional input to the output machinery, and many of the underlying molecular principles, makes the Lap system one of the best-understood c-di-GMP signaling systems to date, and a potent model system to study mechanistic questions. In addition, considering the prevalence of the intracellular HAMP-GGDEF-EAL module (currently 1296 annotated genes in Pfam) and extracellular PAS-like output domain in LapD-like c-di-GMP receptors, as well as in active enzymes for environmental sensing, the mechanism underlying LapD function will be broadly relevant for understanding bacterial transmembrane signaling. Our previous work elucidated the mechanistic underpinnings and regulation of the cytoplasmic portion of LapD (*Navarro et al., 2011*). Here, we focus on the periplasmic events and how they are coupled to the cytoplasmic control elements in the context of the global, c-di-GMP-dependent switching mechanism of the full-length, transmembrane receptor. In particular, we present an activation model for inside-out signaling via LapD based on structures of an unliganded and LapG-bound output domain, and further informed by protein and cell-based switching assays. The results shed light on the control of bacterial proteolysis in the periplasm via conserved signal transduction modules.

## Results

### Preface to the structural studies

We previously identified conserved LapD/LapG-containing operons in many, diverse bacterial species (*Navarro et al., 2011*; *Newell et al., 2011a*). In addition, we reported the crystal structures of the isolated periplasmic output domain of *P. fluorescens* LapD and the LapG protease ortholog from *Legionella pneumophila* (*Navarro et al., 2011*; *Chatterjee et al., 2012*). The former structure revealed an unusual domain-swapped, V-shaped dimer resembling PAS domains (*Figure 1B*), yet several questions remained unanswered. While we identified a surface-exposed loop that contains a strictly conserved GWxQ motif important for LapG binding (*Navarro et al., 2011*), how access of LapG to the loop is controlled was not apparent. Also, the structure of LapG revealed a transglutaminase-like fold with functionally important calcium ions bound near its active site (*Chatterjee et al., 2012*), but did not explain how this protease interacts with LapD. To address these shortcomings, we report here two structures, that of the output domain of a LapD ortholog from *L. pneumophila*, also known and referred to here as CdgS9 (*Levi et al., 2011*), and that of LapG in complex with the same output domain. The comparison of the apo- and LapG-bound states elucidates the periplasmic switching mechanism.

### Structure of the periplasmic output domain of CdgS9

The structure of *L. pneumophila* CdgS9$^{Output}$ (residues of 22–152) was determined from data collected on a crystal grown from selenomethionine-derivatized protein by using single anomalous dispersion (SAD) phasing (see 'Material and methods', and *Table 1* for details). The asymmetric unit contains one

**Table 1.** Data collection and refinement statistics

| | CdgS9[Output] (SeMet) | LapG/CdgS9[Output] (SeMet) | LapG/CdgS9[Output] (native) |
|---|---|---|---|
| Data collection[a] | | | |
| Space group | $P6_122$ | $P2_1$ | $P2_1$ |
| Unit cell axes | | | |
| a, b, c (Å) | 61.58, 61.58, 147.53 | 59.08, 74.69, 62.88 | 75.85, 73.67, 88.22 |
| α, β, γ (°) | 90.0, 90.0, 120.0 | 90.0, 101.8, 90.0 | 90.0, 92.9, 90.0 |
| Resolution Limits (Å) | 30–2.14 (2.22–2.14) | 30–0.29 (2.37–2.29) | 44–2.10 (2.17–2.10) |
| Unique Observations | 9611 (924) | 22,884 (2231) | 55,783 (4959) |
| Completeness | 98.6 (97.6) | 93.8 (92.2) | 97.7 (87.3) |
| Multiplicity | 21.9 (18.6) | 7.2 (5.4) | 4.3 (3.2) |
| Average I/σ | 35.4 (10.5) | 20.1 (2.3) | 12.4 (2.8) |
| $R_{sym}$ (%) | 7.7 (30.9) | 8.9 (67.1) | 8.2 (38.7) |
| Refinement | | | |
| $R_{cryst}/R_{free}$ (%) | 21.9/24.3 | | 17.0/21.1 |
| No. protein molecules | 1 | | 6 |
| No. protein residues | 131 | | 886 |
| No. water molecules | 90 | | 677 |
| Total number atoms | 1158 | | 7826 |
| rmsd bond angles (°) | 1.02 | | 1.05 |
| rmsd bond lengths (Å) | 0.004 | | 0.004 |
| <B> protein (Å$^2$) | 41.2 | | 34.0 |
| <B> water (Å$^2$) | 59.1 | | 38.8 |
| <B> calcium (Å$^2$) | N/A | | 35.3 |
| Ramachandran Plot (%) | | | |
| Favored | 98 | | 98 |
| Outliers | 0 | | 0 |
| PDB code | 4U64 | | 4U65 |

[a]Numbers in parentheses correspond to values in the highest resolution.

molecule of CdgS9[Output], which adopts a PAS-like domain fold with the highest structural homology to CitA (**Reinelt et al., 2003**). A potential biologically relevant dimer involves a crystallographic symmetry mate, burying 3020 Å$^2$ surface area at the twofold symmetry axis via favorable interactions (estimated $\Delta G^{int}$ = −20.2 kcal/mol) (**Krissinel and Henrick, 2007**) (**Figure 1B**, bottom right panel). The residues at the CdgS9[Output] homo-dimer interface tend to be more conserved than residues outside of this region, based on an alignment of 18 putative LapD orthologs from different bacterial species (**Figure 1C**; **Navarro et al., 2011**).

The protomers form a parallel dimer, with the termini lining up at one end of the complex, consistent with their connectivity to the transmembrane helices that flank the periplasmic domain. There is only a minor entanglement of the terminal short β-strands, which connect to the transmembrane helices (**Figure 1B**). Such an arrangement is in stark contrast to our previously determined structure of *P. fluorescens* LapD[Output] (**Figure 1B**, left panel), in which the PAS domain dimer originated from an unusually extensive domain-swap. Although the folds of the dimer half-sites between the *P. fluorescens* LapD[Output] and CdgS9[Output] are almost identical (rmsd of 1.9 Å) and the conserved GWxQ motifs, which are important for LapG binding, are positioned in identical, surface-exposed loops within each half-site, the dimer topology of the two output domain ortholog structures is substantially different (**Figure 1B**). In addition, one notable difference between the crystallographic models for the output domains of LapD and CdgS9 is the relative position of the conserved GWxQ motifs. CdgS9[Output] adopts

a 'parallel' conformation as opposed to the open, V-shaped form that we observed in case of LapD[Output]. As a consequence, the tryptophan residues (W[125] in *P. fluorescens* LapD; W[126] in *L. pneumophila* CdgS9) that form the distal tip of the PAS fold and appear to be a main anchor point for LapG binding (*Navarro et al., 2011*) are farther apart in the *P. fluorescens* dimer than the *L. pneumophila* LapD ortholog (50 Å vs 35 Å between their respective Cβ positions) (*Figure 1B*).

Considering the discrepancy in the two output domain structures, we developed a structure-guided assay, which allows us to distinguish between the two topologies in the context of the intact, full-length LapD protein from *P. fluorescens* (*Figure 2*). The approach relies on the site-specific incorporation

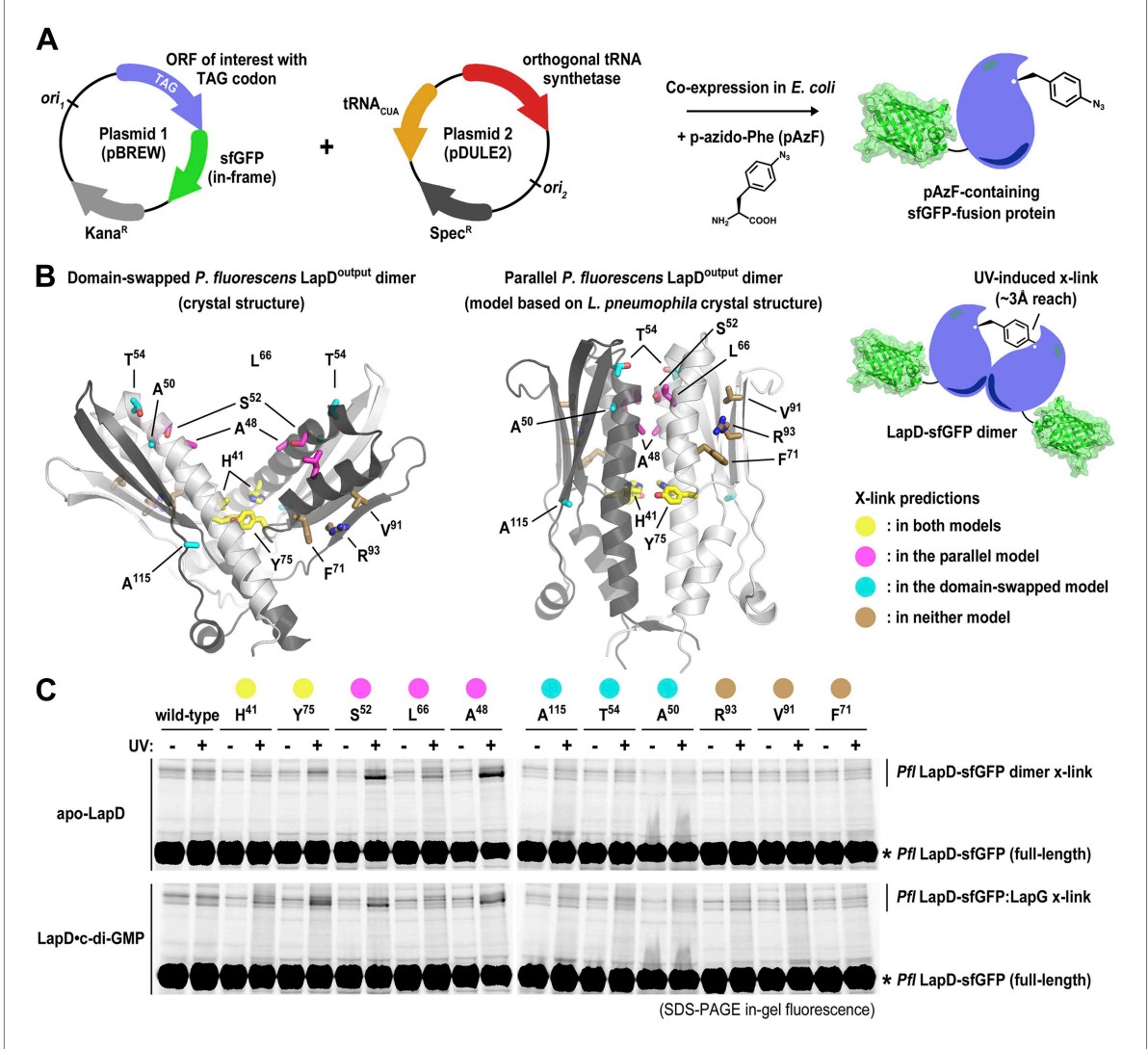

**Figure 2**. Distinguishing between the 'open' domain-swapped and 'closed' parallel conformations of the output domain of LapD. (**A**) Non-natural amino acid incorporation into protein. The UV-photo-activatable non-natural amino acid *para*-azido-phenylalanine (pAzF) was site-specifically incorporated into the output domain of full-length *P. fluorescens* LapD, which was fused C-terminally to superfolder GFP (sfGFP), using a two-plasmid amber-suppression expression system. (**B**) Cross-linking strategy based on structural models. Two different models of the output domain of LapD are shown with the eleven sites of pAzF incorporation shown as sticks and color-coded according to structure-based cross-linking predictions. The left panel shows the domain-swapped, open conformation determined previously (PDB 3PJV; *Navarro et al., 2011*), while the right panel shows a homology model of the 'closed', parallel conformation based on the crystal structure of CdgS9[Output]. (**C**) Crosslinking patterns support a 'closed', parallel model. Cells expressing full-length LapD fused to sfGFP with pAzF incorporated at the indicated positions were harvested by centrifugation and lysed by sonication. Membranes were isolated by ultracentriguation and resuspended in buffer lacking detergent supplemented with and without c-di-GMP. After equilibration, half of each sample was exposed to short-wave UV light. These membrane suspensions were immediately solubilized in 2% SDS and subjected to SDS-PAGE analysis. Gels were imaged by fluorescence. As indicated, cross-links appear as band-shifts with a higher molecular weight compared to LapD-sfGFP.

of a non-natural amino acid with altered functionality, in this case *para*-azido-phenylalanine (pAzF) for target cross-linking (*Figure 2A*; *Chin et al., 2002*; *Mehl et al., 2003*; *Peeler and Mehl, 2012*). Upon UV irradiation, pAzF reacts efficiently with N-H and C-H bonds in its vicinity, forming covalent adducts within a radius of approximately 3 Å (*Keana and Cai, 1990*). Specifically, we introduced pAzF via an amber codon-suppression system at sites that we predict would yield cross-links either in only one of the two topologies ($A^{50}$, $T^{54}$, $A^{115}$, based on V-shaped *P. fluorescens* model; $A^{48}$, $S^{52}$, $L^{66}$, based on a homology model using the parallel *L. pneumophila* structure as the target), in both ($H^{41}$, $Y^{75}$) or in none ($F^{71}$, $V^{91}$, $R^{93}$) (*Figure 2B*). Choosing several sites in each category increases confidence in the results. LapD variants containing pAzF incorporated at these positions were expressed in a construct wherein the protein was fused to super-folder green fluorescent protein (sfGFP) (*Pedelacq et al., 2006*) at its C-terminus (*Figure 2A*), which enables the use of in-gel fluorescence to detect monomeric and cross-linked species in SDS-PAGE with high specificity and sensitivity using crude lysates or partially purified proteins. In this particular case, we expect to observe a second band corresponding to a covalent LapD-sfGFP dimer only if the pAzF residue was incorporated at or very close to the dimerization interface.

We expressed each full-length *P. fluorescens* LapD variant carrying a site-specific pAzF residue in *Escherichia coli*, isolated the membrane fraction, and exposed half of the respective samples to UV, the other half was processed identically but without excitation of pAzF. Proteins were separated in SDS-PAGE, and fluorescent bands were detected using a gel imager. In the absence of UV light, no cross-linked products were observed. Also, no cross-links occurred in the wild-type protein lacking pAzF. Notably, no cross-linked LapD dimers were observed when pAzF was introduced at positions that would support dimerization in the crystallographic domain-swapped, V-shaped LapD$^{Output}$ model, either in the absence or presence of c-di-GMP. The only cross-linked LapD dimers were seen with proteins that contained pAzF at interfacial positions that were consistent with both models ($H^{41}$, $Y^{75}$) or identified in the parallel, unswapped model (e.g., $A^{48}$, $S^{52}$, $L^{66}$) based on the CdgS9$^{Output}$ structure. Taken together, the cross-linking data clearly indicate that the topology observed in the CdgS9$^{Output}$ structure represents the biologically relevant model, which is conserved in the two LapD orthologs, LapD and CdgS9, despite only moderate overall sequence conservation (31% identity/52% similarity). We suspect our previously published domain-swapped model of the *P. fluorescens* output domain was an artifact of expression and/or crystallization.

## Structure of a LapG–LapD$^{Output}$ complex

A deeper understanding of the regulation of periplasmic proteolysis via LapD requires a detailed structural model of a LapG–LapD$^{Output}$ complex. We previously showed that the isolated output domain of *P. fluorescens* or *L. pneumophila* LapD orthologs binds its corresponding LapG protease counterparts readily (*Navarro et al., 2011*; *Chatterjee et al., 2012*). In addition, we demonstrated that the mode of interaction is conserved since we also could generate mixed complexes between the output domain from one strain and the protease of the other, and these interactions relied on the presence of the surface-exposed tryptophan residue ($W^{125}$ in *P. fluorescens* LapD, $W^{126}$ in *L. pneumophila* CdgS9) in the conserved GWxQ loop of LapD (*Chatterjee et al., 2012*). While the homogeneous complexes failed to crystallize thus far despite the best of our efforts, we were successful in determining the structure of *L. pneumophila* CdgS9$^{Output}$ bound to *P. fluorescens* LapG (see details in 'Materials and methods') (*Figure 3*). The use of orthologous proteins as surrogates is common practice in obtaining sought-after crystal structures, and our validation experiments (see below) attest to the conservation of key features and the functional relevance of these structure-based models.

The crystallized complex consists of two CdgS9 output domain molecules that adopt a similar parallel dimer conformation as in the apo-CdgS9$^{Output}$ structure (*Figures 1B and 3A*). The output domain dimer is bound to one LapG molecule at a lateral docking site (*Figure 3A*), with an interaction surface of 1290 Å$^2$ (with a calculated $\Delta G^{int} = -16.3$ kcal/mol; PISA [*Krissinel and Henrick, 2007*]) and involving both molecules of the CdgS9$^{Output}$ dimer to a similar extent (*Figure 3B*). As predicted by the biochemical analysis (*Navarro et al., 2011*; *Chatterjee et al., 2012*), the aforementioned conserved tryptophan residue ($W^{126}$) is the major anchor point on CdgS9 as it inserts into a hydrophobic pocket at the bottom of LapG (*Figure 3A*, right panel). The pocket is formed by strand β4 of LapG's C-terminal lobe and several other disjointed motifs preceding β4. While there are contributions from a limited number of hydrogen bonds and a water-mediated interaction (*Figure 3A*), the interface appears to be dominated by hydrophobic contacts and shape-complementarity of the interacting elements. Besides the

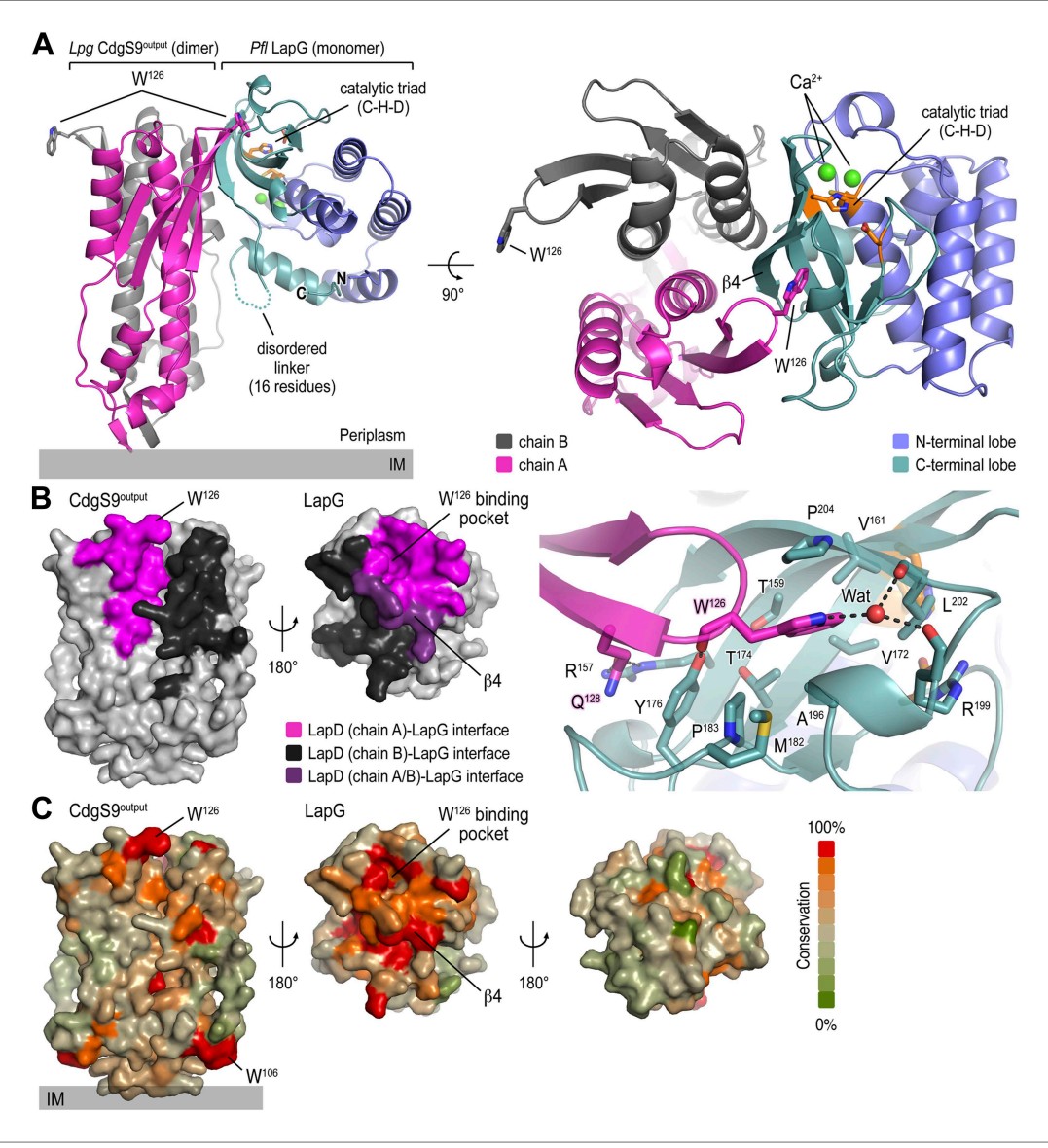

**Figure 3**. Crystal structure of a *P. fluorescens* LapG and *L. pneumophila* CdgS9^Output complex. (**A**) Overview of the complex structure. The LapG-CdgS9^Output complex is shown as a ribbon representation, with the two protomer chains of CdgS9^Output colored in pink and gray, and the N- and C-terminal lobes of LapG shown in slate and cyan, respectively. The conserved catalytic triad (cysteine-histidine-aspartate; C-H-D) is shown as sticks with the carbon atoms in orange. The highly conserved tryptophan residue (W^126) in each protomer of CdgS9^Output is presented as sticks, calcium ions of LapG are shown as green spheres. The relative position of the inner cell membrane (gray bar) is indicated. Two orthogonal views are shown. (**B**) Interaction interface between CdgS9^Output and LapG. The interface footprints on LapG and the CdgS9 half-sides (pink, purple and black) were highlighted on the accessible surface area of the individual proteins (left panel). The binding pocket for CdgS9 W^126 in LapG is shown as a close-up view (right panel). A 180° rotation was applied to LapG from the complex structure to view the interface. (**C**) Surface conservation of CsgS9^Output and LapG interaction interface. Based on the alignment of 18 sequences of LapD and LapG orthologs, the sequence conservation was mapped onto the accessible surface of each protein. The surface is colored according to the degree of sequence conservation. For LapG, two views, separated by a 180° rotation, are shown.

binding pocket for the conserved tryptophan residue, LapG contributes surfaces surrounding the pocket and the central strand β4 that binds to the CdgS9^Output dimer interface, yielding a continuous interfacial area bridging both protomers of the CdgS9^Output dimer.

Based on an alignment of 18 distantly related LapDG ortholog pairs (*Navarro et al., 2011*), we mapped sequence conservation onto the molecular surface of CdgS9$^{Output}$ and LapG. Notably, there is a disparity regarding surface conservation between the two proteins: LapG presents a cluster of conserved residues on the side that coincides with the active site and the region that interacts with CdgS9$^{Output}$, while the opposite side of the protease shows poor sequence conservation (*Figure 3C*, right panel). In contrast, CdgS9$^{Output}$ is less conserved overall (37% identical to *P. fluorescens* LapD$^{Output}$) and shows only a moderate clustering of conserved residues (*Figure 3C*, left panel). Yet, the main docking site for the CdgS9–LapG interaction, the loop that contains the GW$^{126}$xQ motif, is well conserved. As discussed below, pronounced shape complementarity between the interacting surfaces may contribute to the stable recruitment of LapG to the periplasmic output domain of LapD orthologs.

## Comparison between apo- and LapG-bound CdgS9$^{Output}$

Determination of the structure of apo- and LapG-bound CdgS9$^{Output}$ permits us to describe the conformational changes that tune the affinity of LapD-like receptors for LapG proteases. Given that the core LapG-binding motif (the GWxQ loop) is surface-exposed in the apo-CdgS9$^{Output}$ structure and is potentially available for engaging LapG, it was important to address how differential LapG binding would be achieved in the two distinct states.

For their comparison, we superimposed the apo-CdgS9$^{Output}$ dimer onto the CdgS9$^{Output}$ domain of the complex (*Figure 4*). The conformation of the individual output domain protomers varies only slightly in the two states (rmsd between 1.1 and 1.7 Å). However, from a top view a change in the output domain dimer interface is apparent (*Figure 4A*, left panel). As a result, a cradle is created, which buttresses strand β4 of LapG while providing additional anchor points via the conserved GWxQ binding loop on one dimer half-side and helix α2 on the other. The CdgS9 helix butts against the β-sheet of the C-terminal LapG lobe, just underneath the protease active site.

This conformational change is realized through the differential packing of the central helices within the output domain dimer (*Figure 4B,C*). In the apo-state, the output domain dimer of CdgS9 contains central four-helix bundles, either via packing of helices α1 and α2 of the PAS domain fold close to the distal end, or helices α1 and α4 at the membrane-proximal region. The types of interactions that facilitate the formation of the two 4-helix bundle segments are quite different. Closer to the membrane, a cluster of 4 phenylalanine residues (2 per protomer; F$^{33}$ and F$^{34}$ of CdgS9) stack in a perpendicular fashion, as opposed to the packing near the distal tip, which involves the burying of two symmetry-related arginine residues (R$^{75}$ of CdgS9) (*Figure 4C*). As a consequence, the overall structure of the output domain appears bulged (*Figure 4B*).

When engaged with LapG (*Figures 3A and 4B*), this perfect symmetry between the two protomers observed in the apo-state is broken. The resulting complex is asymmetric with regard to the output domain dimer, manifested by a 5 Å tilt and vertical offset of one PAS domain fold relative to the other. The central helices α1 are pushed closer together—a movement permitted by smaller residues at the interface (S$^{48}$ and S$^{52}$), a perpendicular-to-parallel stacking transition of one of the F$^{33}$–F$^{34}$ residue pairs, and the relocation of R$^{75}$ from the dimer interface to a more surface-exposed position. The net result is a transition from a four-helix bundle with poor shape complementarity with LapG to a two-helix bundle and an overall shape that is able to accommodate LapG (*Figure 4C*). We argue that the isolated output domain is in a deregulated state and spontaneously populates the high-affinity conformation for LapG binding (on-state) since regulatory domains that would lock the receptor in its off-state (the apo state) or on-state are lacking. This conclusion is consistent with the observation that the isolated output domain is an inefficient competitor for the interaction between LapG and full-length LapD (data not shown). Nevertheless, the conservation of the interfaces and motifs involved in LapG binding in the otherwise fairly divergent LapD family of receptors suggest that the proposed mechanism is more generally applicable.

Taken together, the crystal structures of apo-CdgS9$^{Output}$ and a CdgS9$^{Output}$-LapG complex and our previous work (*Newell et al., 2009*; *Navarro et al., 2011*; *Newell et al., 2011a*) yield a detailed model for the activation of LapD by c-di-GMP binding to its cytosolic module. It involves the release of an autoinhibitory interaction, triggering a conformational change in the periplasmic domain through the juxtamembrane HAMP domain and transmembrane segments. High-affinity LapG binding to the off-state of LapD is prevented by an apparent surface mismatch. In contrast, activated LapD displays a new surface that is cryptic in the apo state, spans both protomers of the output domain dimer, and is optimized for interaction with LapG (*Video 1*).

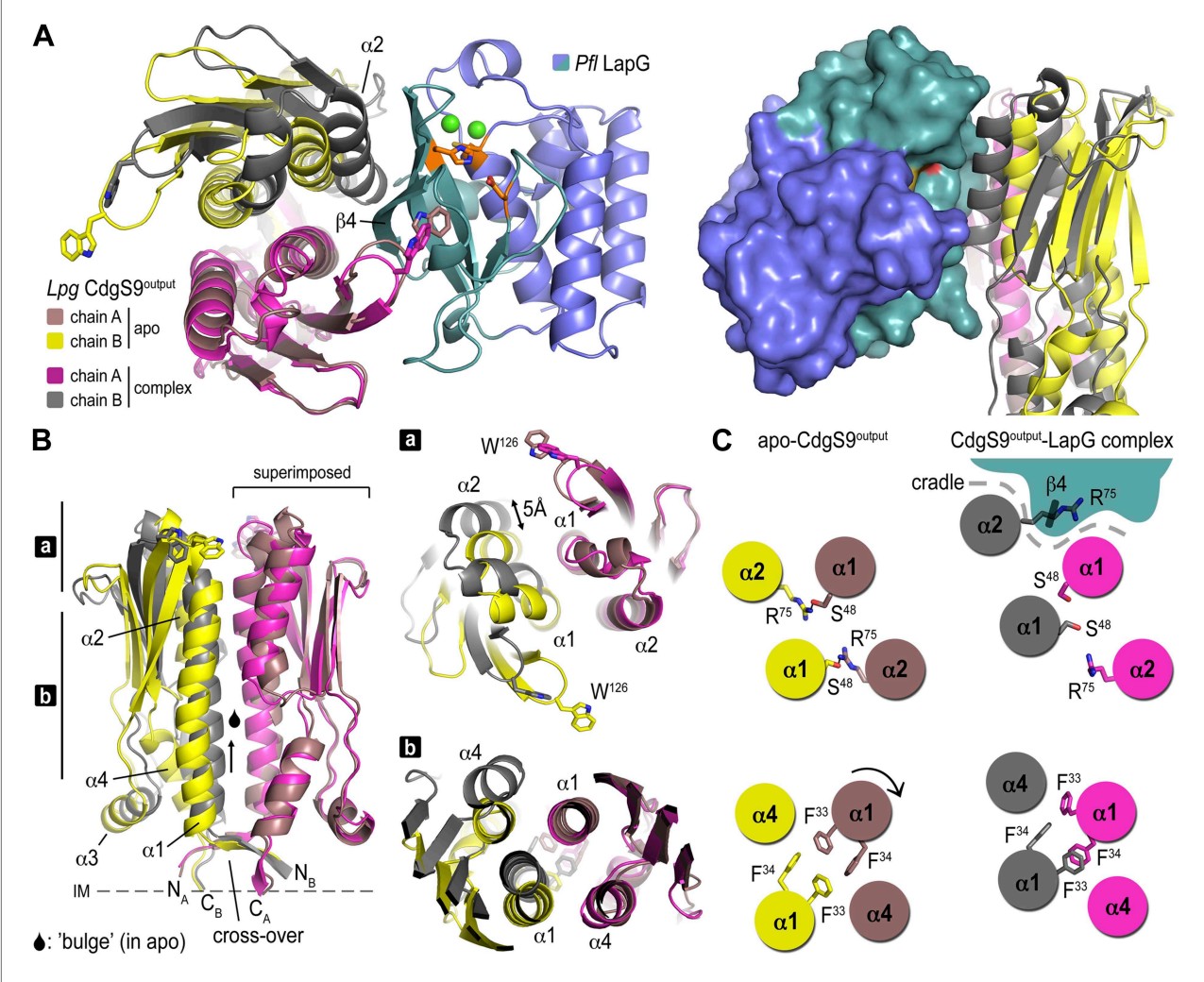

**Figure 4**. Conformational changes in CdgS9$^{Output}$ between the apo-state and in complex with LapG. (**A**) Overview of the superposition of apo-CdgS9$^{Output}$ and LapG-bound CdgS9$^{Output}$. Chain A of the CdgS9$^{Output}$ dimer was used as the reference for the superposition. Chains A and B of apo-CdgS9$^{Output}$ are colored tan and yellow, while chains A and B of LapG-bound CdgS9$^{Output}$ are colored pink and gray, respectively. The N- and C-terminal lobes of LapG are colored in slate and cyan, respectively. Both cartoon (left panel) and surface (right panel) representations of LapG are shown. (**B**) Components of the conformational change. The superposition in (**A**) is shown as a side-view, depicting the helical motions of α1, α2, and α4. Two perpendicular views are shown, with the right panel showing slices of the top view. (**C**) Cartoon model. The structural transition from an apo-state (yellow/tan) to a LapG binding-competent state (pink/gray) of CdgS9$^{Output}$ based on the analysis shown in panels (**A**) and (**B**) is illustrated. The top pair depicts changes at the distal tip of the output domain; the bottom pair illustrates changes in the membrane proximal region of the output domain.

## Structure of *P. fluorescens* LapG in the CdgS9$^{Output}$–LapG complex

Previously, we determined the crystal structure of calcium-bound *L. pneumophila* LapG (*Chatterjee et al., 2012*), a protease with transglutaminase fold (*Ginalski et al., 2004*). Here, we obtained a molecular model of the CdgS9-bound ortholog from *P. fluorescens*. In order to ascertain the degree of conservation of the two LapG protease orthologs and to detect any potential conformational changes that may occur upon binding to a LapD ortholog, the two structures were superimposed and analyzed for differences (***Figure 5A***). Despite moderate sequence conservation between the two orthologs (40% identity/54% similarity), the two structures are very similar with regard to their overall fold (Cα rmsd of 1.09 Å over 136 atoms). However, there are a few significant differences between the two structures. First, we identified a rare di-calcium binding motif in the N-terminal lobe of *P. fluorescens* LapG, which is in contrast to the single calcium ion bound at the same site in the *L. pneumophila* LapG structure (***Figure 5B***). Yet, the two orthologs share the same conserved coordinating residues required for di-calcium binding (***Figure 5C***)

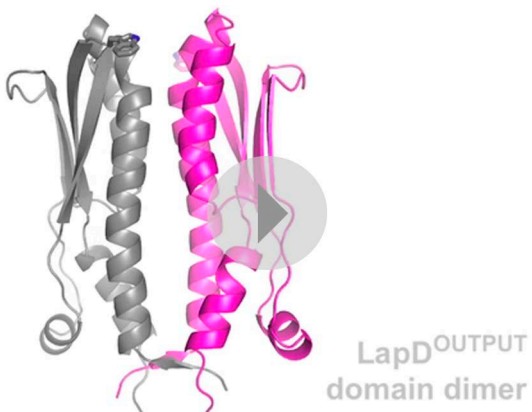

LapD^OUTPUT domain dimer

**Video 1**. Composite structural overview. The video shows several interpolations between the apo-state and the LapG-bound output domain dimer of CdgS9, based on the crystallographic data presented here. It depicts the conformational changes between the two states of the periplasmic output domain and how they lead to LapG recruitment.

(*Chatterjee et al., 2012*). Additional experiments will be necessary to ascertain the functional purpose of these alternate calcium-binding modes; however, it is tempting to suggest these differences help tune their respective substrate specificities given the proximity of the calcium atoms to the catalytic triad.

The second conformational change observed between CdgS9^Output-bound *P. fluorescens* LapG and the *L. pneumophila* LapG ortholog localizes to the site of interaction with LapD orthologs, which involves the peeling away of β5 upon binding to LapD, whereas β5 and part of the linker are ordered in the stand-alone LapG structures (*Figure 5A*). Notably though, even in these structures this region appears to be more flexible than the core of the protease based on an analysis of their corresponding temperature factors and the presence of largely disordered segments in its vicinity.

## Validation of the complex structure as a model for LapD–LapG interactions

Undoubtedly there are concerns when working with a mixed complex containing isolated domains of proteins from different bacterial genera, foremost the validity of the crystallographic models and their relevance for LapD function. This issue is particularly important considering that intact LapD is a transmembrane protein with a cytoplasmic module that is crucial for the regulation of signal transmission across the membrane. To assess the soundness of our structure-based models, we again turned to a cross-linking assay utilizing non-natural amino acid pAzF incorporation. The cross-linking assay awards several advantages over other binding experiments. It provides site-specific information that can be directly correlated with crystallographic data. In addition, it is very sensitive, allowing us to work at low, more physiologically relevant protein concentrations that support c-di-GMP-dependent regulation of LapD, which is lost at higher protein concentrations required in alternative binding assays. Also, the assay reflects on equilibrium binding. Finally, although the cross-linking efficiency is low, the results are semi-quantitative since empirically the fraction of the successfully established cross-links appears to be proportional to the number of bound molecules. We followed the strategy that we employed to assess LapD homo-dimerization (*Figure 2*), but in contrast to previous experiments, we introduced pAzF site-specifically into sfGFP-tagged LapG and assessed UV-induced cross-linking to LapD variants, which can be detected as up-shifts of a fluorescent band in SDS-PAGE gels (*Figure 6A*).

Specifically, we introduced pAzF at positions in *P. fluorescens* LapG that are close to the crystallographic LapG-CdgS9^Output interface (A[169], Y[176], V[205], or Y[206]) or at a site distal to the interfacial region (Y[108]) (*Figure 6B*). The purified pAzF-containing, sfGFP-tagged *P. fluorescens* LapG variants were mixed with (i) *L. pneumophila* CdgS9^Output to validate the crystallographic complex; (ii) *P. fluorescens* LapD^Output to assess the conservation of interactions in a homologous complex; and (iii) full-length, c-di-GMP-activated *P. fluorescens* LapD, which will enable us to extend results to the native, regulated receptor. We also included a control where we do not add any LapD variant, which will allow us to distinguish between LapG–LapD interactions and possible LapG homo-dimers (*Figure 6B*). Without UV irradiation, no cross-links were observed in any of the conditions. We also did not observe any band shift in the sample that only contained pAzF-derivatized LapG variants but no LapD, consistent with our previous studies that did not indicate signs of LapG oligomerization (*Chatterjee et al., 2012*). In addition, pAzF at position Y[108] of LapG, which is distal to the LapD–LapG interface, did not support cross-linking. In contrast, all the LapG variants that contained a pAzF residue at positions that, based on the structure, are predicted to be close to the interaction site yielded band shifts to variable extents (V[205]pAzF > Y[206]pAzF/Y[176]pAzF/A[169]pAzF >> Y[108]pAzF) (*Figure 6B*). It is worth noting that the isolated output domains of *L. pneumophila* CdgS9 and *P. fluorescens* LapD cross-link to LapG readily, suggesting that they are in a binding-competent state, which is in agreement with our crystallographic analysis.

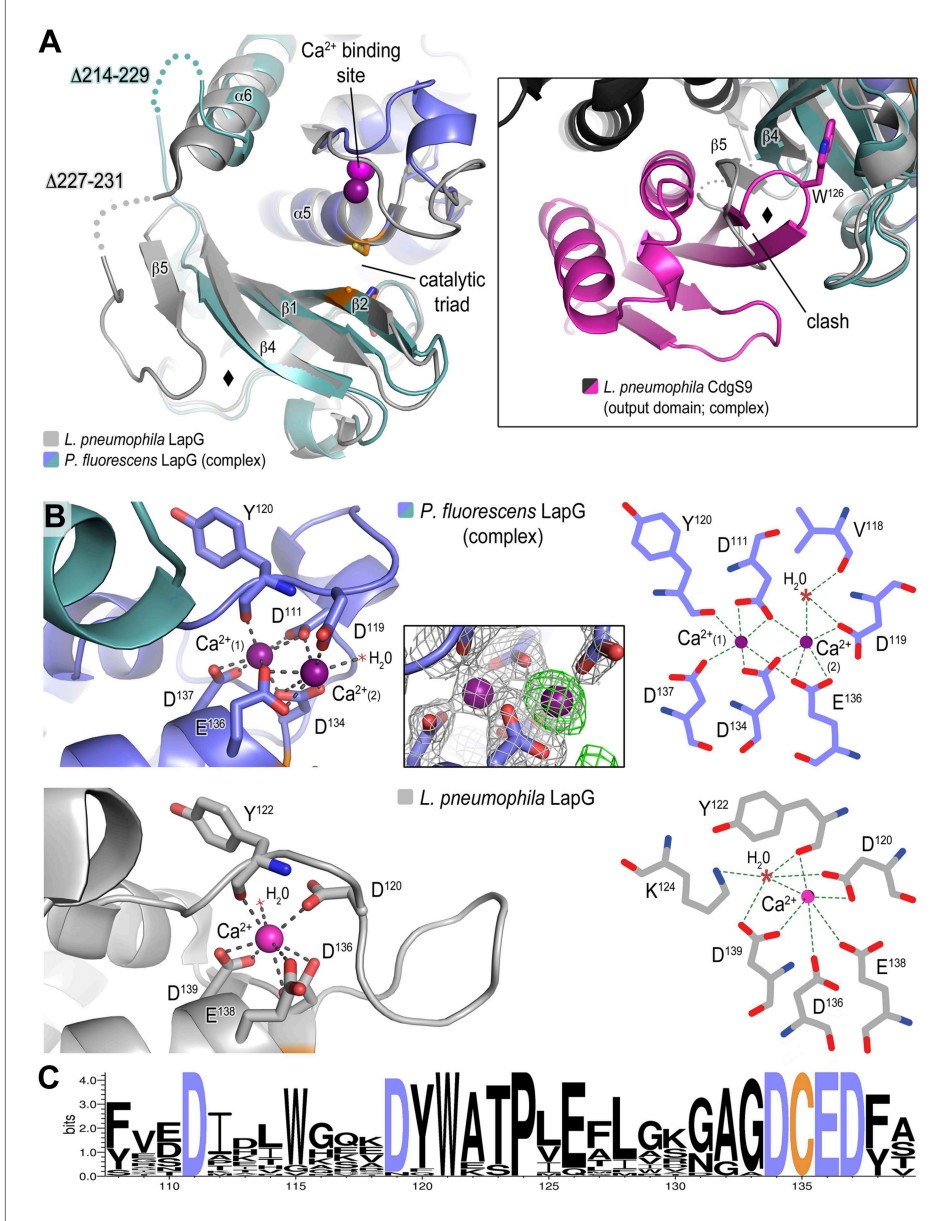

**Figure 5**. Structure of *P. fluorescens* LapG in complex with CdgS9[Output]. (**A**) The structure of *L. pneumophila* LapG shown in gray (PDB: 4FGO, *Chatterjee et al., 2012*) was superimposed on that of *P. fluorescens* LapG, colored in slate and cyan, when in complex with CdgS9[Output]. A close-up view of the differences observed in the C-terminal lobe of the two LapG structures is shown (right inset). Disordered regions not resolved in the crystal structures are shown as dots. Calcium ions are represented as magenta spheres. The catalytic triad residues (C-H-D) are shown as sticks with the carbon atoms colored in orange. (**B**) The Ca[2+] binding sites of LapG. *P. fluorescens* (top panel, blue cartoon) and *L. pneumophila* LapG (bottom panel, gray cartoon) coordinate two or one calcium ion, respectively. In *P. fluorescens* LapG, an $F_o$–$F_c$ omit map contoured at 3σ (green mesh, inset of top panel) shows strong evidence for a second calcium ion not seen in the *L. pneumophila* structure, which is made possible by local conformational differences in the so-called calcium binding loop that bring D[111] of *P. fluorescens* into position to coordinate the additional calcium ion. A LIGPLOT (*Wallace et al., 1995*) (right panels) depicts the coordination of calcium ions. (**C**) Conservation of the di-calcium binding site. A WebLOGO plot (*Crooks et al., 2004*) generated from an alignment of 18 LapG ortholog sequences demonstrates the conservation of coordinating residues (blue font) as well as the nucleophilic cysteine (yellow font).

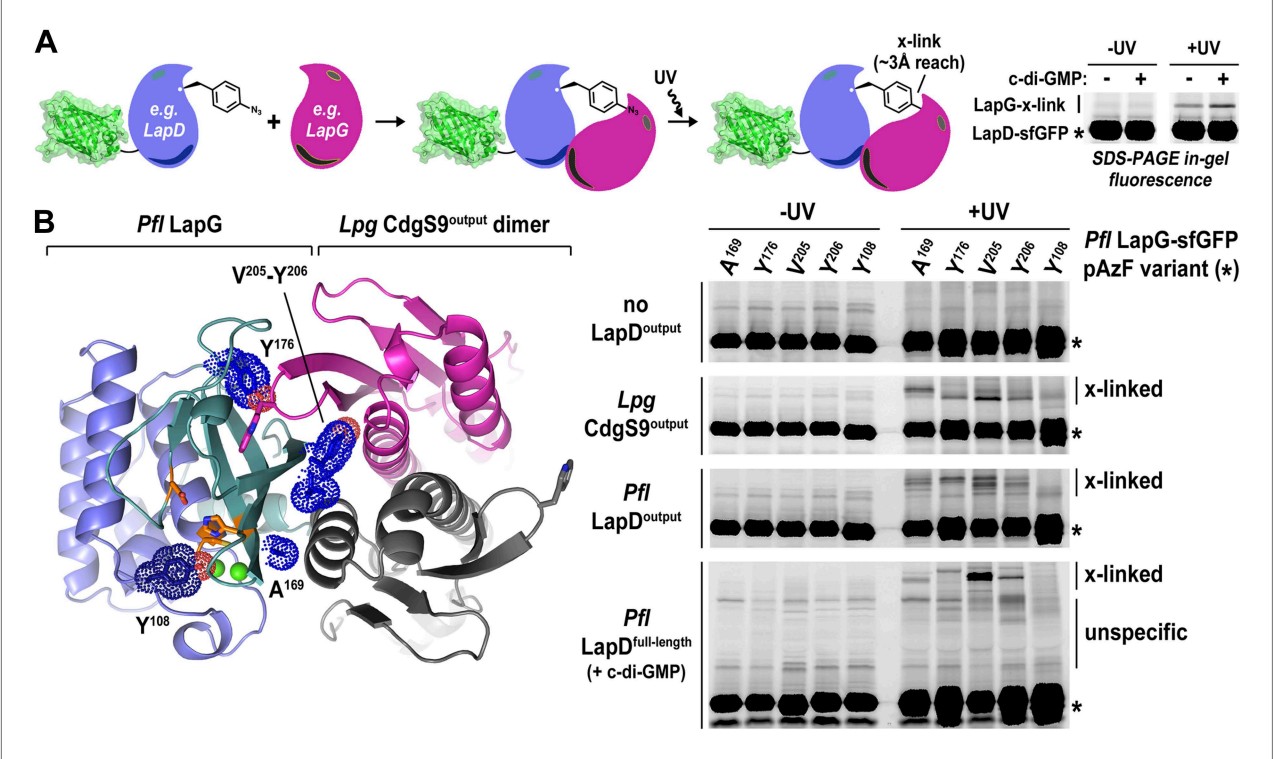

**Figure 6**. Validation of the crystallographic CdgS9output-LapG complex interface. (**A**) Schematic overview of the Lapd-LapG cross-linking assay. The right panels show a representative experiment in which detergent-solubilized LapD-sfGFP is incubated with non-fluorescent LapG-containing pAzF incorporated at a site predicted to be at the LapD–LapG binding interface in the presence and absence of c-di-GMP. Upon UV irradiation, a fluorescent band corresponding to LapD-sfGFP covalently linked to LapG is observed, which is more intense in the presence of c-di-GMP indicative of more LapD–LapG binding. Other experiments were carried out with a LapG-sfGFP/LapD pairs. (**B**) Structure-guided cross-linking of LapD variants and LapG. The complex structure (left panel) highlights four sites on LapG (A169, Y176, V205, and Y206) that span the width of the binding interface plus an additional fifth site (Y108) that is not at the interface in which pAzF was incorporated. Each of these five LapG-sfGFP pAzF-containing variants were purified and incubated with purified non-fluorescent CdgS9output, LapDoutput and c-di-GMP activated, detergent-solubilized full-length *P. fluorescens* LapD prior to irradiation with UV light. Samples were analyzed by SDS-PAGE and gels imaged by fluorescence as described in **Figure 2**.

However, cross-linking is more robust in the presence of c-di-GMP-activated full-length LapD protein (**Figure 6A**). These data are consistent with the conclusion that our structural models and assays accurately report on the complex formation between full-length LapD and LapG.

## Stoichiometry of the complex

Next, we sought support for the asymmetric ensemble of a CdgS9output dimer bound to one LapG protease observed in the crystal structure of the complex. To assess whether the asymmetric 2:1 complex observed in the crystal structure was compatible with the binding of a second LapG molecule, we used the CdgS9output protomer with the unliganded W126 residue (CdgS9B; **Figure 7A**) as a reference in a superposition of the output domain protomer that engages LapG involving the conserved tryptophan residue. From this analysis, it is clear that the potential second binding site does not allow as tight a binding as the crystallographically determined docking site due to sub-optimal shape complementarity (**Figure 7A**, note gap indicated in bottom panel; statistic shape complementarity (**Lawrence and Colman, 1993**; **Collaborative Computational Project N, 1994**) of 0.72 for the crystallographic interface vs 0.31 for the modeled, second binding site). This observation is consistent with the asymmetric distribution of conformational changes at the output domain dimer interface (see above; **Figure 4A**).

Further experimental evidence is provided by SEC-MALS experiments that report the absolute molecular weight of proteins and complexes as they elute from a size-exclusion chromatography column (**Figure 7B**). The purified protein complex that was derived from co-expression of its components and yielded the native complex crystals elutes in a single peak, with an experimentally determined average

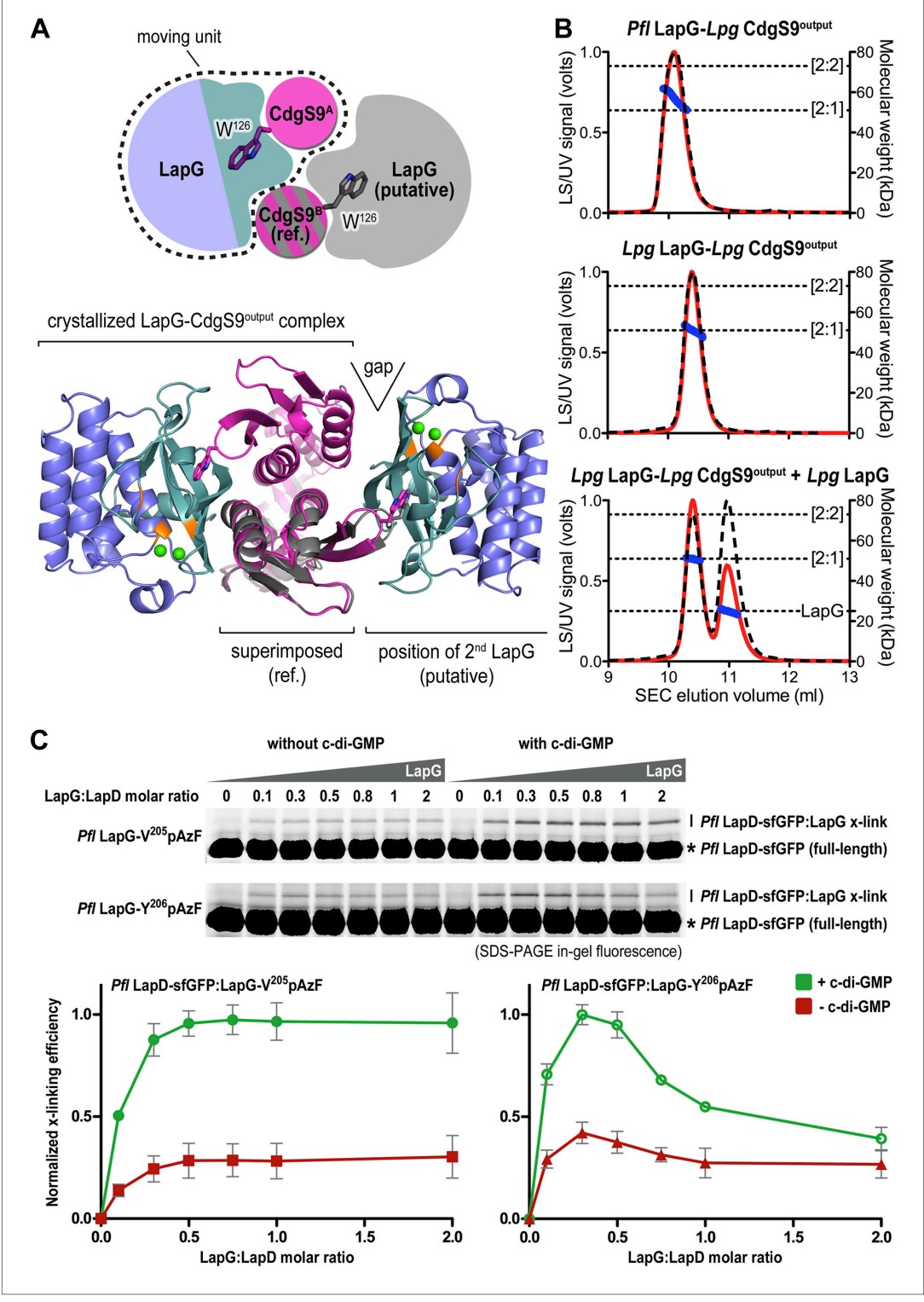

**Figure 7**. Stoichiometry of the LapD–LapG complex. (**A**) Structural model of a 2:2 CdgS9$^{Output}$–LapG complex. The CdgS9$^{Output}$ domain that engages LapG via its GW$^{126}$xQ loop was superimposed onto the output domain of the complex structure, which is not bound to LapG. The open arrow indicates poor shape complementarity at the putative second LapG binding site. (**B**) Stoichiometry of CdgS9$^{Output}$-LapG complex in solution. SEC-MALS was used to determine the absolute molecular weight of the CdgS9$^{Output}$-*P. fluorescens* LapG complex used for crystallization (top panel), a co-expressed complex of *L. pneumophila* LapG bound to CdgS9$^{Output}$ (middle panel), and the latter

*Figure 7. Continued on next page*

*Figure 7. Continued*

complex incubated with an excess *L. pneumophila* LapG (bottom panel). Red traces and dashed lines show the signal of the light scattering and UV detector, respectively. Dotted, vertical lines indicate the theoretical molecular weight of the 2:1 and 2:2 output domain-protease complexes based on their primary sequence. Blue points across the elution peaks refer to the calculated molecular weight as the proteins elute from the gel filtration column. (**C**) Assessment of LapG binding to full-length LapD by covalent cross-linking. Purified, non-fluorescent LapG possessing pAzF at positions $V^{205}$ (top gel) or $Y^{206}$ (bottom gel) was titrated into a fixed quantity of detergent solubilized LapD-sfGFP in the absence (left side of gel) or presence (right side of gel) of c-di-GMP (20 µM) and allowed to incubate for 20 min. Samples were irradiated with UV light for 5 min and analyzed by SDS-PAGE. Band intensities of LapD-sfGFP cross-linked to LapG from three independent experiments were measured, normalized to non-crosslinked LapD-sfGFP, averaged, and plotted as function of the LapG:LapD molar ratio (±SD), which showed maximum crosslinking at ~0.3–0.5 LapG molecules to 1 LapD molecule (bottom panels).

molecular weight of 56.9 kDa (theoretical molecular weight for a CdgS9$^{Output}$ dimer bound to one LapG molecule: 52.3 kDa). Investigation of the measurements across the peak reveals a trend towards slightly higher molecular weights at the beginning of the elution peak, but it never reaches values that would indicate the formation of a stable 2:2 complex (two LapG molecules bound to a CdgS9$^{Output}$ dimer; theoretical molecular weight of 75.2 kDa). We also purified the homogeneous complex with both LapG and CdgS9$^{Output}$ coming from *L. pneumophila*. This sample also yielded a single peak with a molecular weight of 50.9 kDa, consistent with a 2:1 complex (a CdgS9$^{Output}$ dimer bound to one LapG molecule; *Figure 5B*, middle panel). The trend towards higher molecular weights was less apparent, attesting to the homogeneity of this complex. More importantly, even addition of an excess *L. pneumophila* LapG to the purified complex failed to drive the formation of a 2:2 complex (*Figure 7B*, bottom panel).

Similar experiments for the *P. fluorescens* proteins could not be performed due to complexity of the LapD$^{Output}$ elution profiles (data not shown), which indicated high polydispersity and the presence of multiple species that complicated the analysis by SEC-MALS. As an alternative, we turned to the site-specific cross-linking assay introduced earlier to assess complex formation between full-length *P. fluorescens* LapD and LapG in a semi-quantitative manner (*Figure 7C*). Here, we titrated sfGFP-tagged full-length *P. fluorescens* LapD with increasing amounts of *P. fluorescens* LapG with pAzF replacing either $V^{205}$ or $Y^{206}$, two positions that supported robust cross-linking but would tether to opposite protomers of the LapD dimer (*Figure 6C*). Based on the crystal structure of the complex, $Y^{206}$pAzF is predicted to cross-link to the LapD protomer that engages LapG with its GWxQ loop, whereas $V^{205}$pAzF faces helix α2 of the other protomer.

Both LapG variants showed poor but detectable cross-linking in the absence of c-di-GMP (*Figure 7C*), consistent with previous pull-down experiments (*Newell et al., 2011a*). Activation of LapD by the addition of c-di-GMP produced more robust cross-linking across the LapG concentration series. Strikingly, in all cases, maximum cross-linking was achieved at sub-stoichiometric LapG concentrations, roughly corresponding to a ratio of one LapG molecule per LapD dimer. While the LapG-$V^{205}$pAzF variant resembled a regular saturation-binding curve, LapG-$Y^{206}$pAzF cross-linking peaked between LapG:LapD molar ratios of 0.3–0.5 and then dropped at higher LapG concentrations (*Figure 7C*). No double-shift was observed at any condition, which would indicate the formation of a 2:2 complex, suggesting that this is a rather rare event.

These data are consistent with the formation of an asymmetric LapD–LapG complex in *P. fluorescens*, but they also suggest distinct modes of binding, an argument that is based on the disparity observed between LapG-$V^{205}$pAzF and LapG-$Y^{206}$pAzF. The difference in cross-linking profiles may be rationalized by the observation that $V^{205}$pAzF could form a covalent bond with the LapD output domain protomer for which binding does not require interaction with the tryptophan-containing loop (*Figures 6B and 8A*). Taken together, these experiments establish the asymmetric complex observed in the crystals as the predominant species in solution and the membrane, and this complex appears to be conserved across the two bacterial genera assessed here.

## Manipulation of the LapD–LapG interaction

Our studies to date have established LapDG as an efficient signaling node to control cell adhesion and biofilm formation (*Newell et al., 2009*; *Navarro et al., 2011*; *Newell et al., 2011a*; *Newell et al., 2011b*). The structures provide not only mechanistic insight but also a blueprint for the development of vehicles that would disrupt the LapG–LapD interaction, and hence facilitate dispersal of biofilms. As

proof-of-concept, we designed peptides that contain the conserved GWxQ binding motif and would be predicted to compete with the native interaction. As a control, we used a peptide with the corresponding W[125]E mutations, which in analogy to the mutant LapD variant reported previously (*Navarro et al., 2011*; *Chatterjee et al., 2012*), should not bind to LapG and hence be a less effective competitor of the LapD–LapG interaction.

Initially, we used the cross-linking assay to assess the effect of the peptides on the *P. fluorescens* LapD–LapG interactions. Purified sfGFP-fused LapD was mixed with LapG variant Y[206]pAzF or V[205]pAzF that had been pre-incubated with either the MDGWMQA or the MDGEMQA peptides. We used both cross-linking positions since they yielded non-equivalent results previously. Based on the crystal structure of the complex, Y[206]pAzF would be predicted to covalently attach to the LapD dimer half-side that engages with LapG via its GWxQ loop, whereas V[205]pAzF points towards the other protomer that uses the interaction surface distal to its GWxQ loop (*Figures 6B and 8A*). In our previous experiments, we observed stronger cross-linking with V[205]pAzF, while Y[206]pAzF showed similar trends but unique cross-linking behavior in the LapG titration experiments, which indicated that the Y[206]pAzF is more sensitive and may report on a more regulated molecular interaction than the V[205]pAzF position.

Here, we demonstrate that increasing amounts of the MDGWMQA peptide, but not of the mutated MDGEMQA, decrease the cross-linking efficiency of Y[206]pAzF (*Figure 8A*). At the highest peptide concentration, cross-linking is reduced to about 60% of the unperturbed interaction. Although such robust effects require a large excess of the competitor, they are specific for the tryptophan-containing peptide. Inhibition is less pronounced when we used the LapG-V[205]pAzF variant (*Figure 8A*, inset). Finally, the addition of the peptide with the native sequence to LapG pull-downs from *P. fluorescens* lysates reduces the amount of LapD coming down with LapG, whereas the mutant peptide does not compete with the native interaction (*Figure 8B*). Together, these data indicate that inhibition of the native LapG–LapD interaction presents a feasible strategy to interfere with this and similar signaling systems.

## Mutational analysis of LapD

Previously, we elucidated intra-molecular interactions in the cytoplasmic module that stabilize an auto-inhibited conformation of LapD. In particular, we showed that the S helix binds to the EAL domain of the same molecule, placing the GGDEF domain on top of the c-di-GMP binding site of the EAL domain, preventing activation at low cytoplasmic c-di-GMP levels. We also identified mutations (S[229]D and F[222]E) that confer a hyper-biofilm phenotype in *P. fluorescens* and fail to respond to the appropriate environmental cue, in this case phosphate availability. S[229]D and F[222]E locate to the S helix and are thought to destabilize its interaction with the EAL domain (*Navarro et al., 2011*). Using the site-specific cross-linking assay, we interrogated the underlying molecular mechanism of these and other, structure-guided mutants in more detail by assessing directly how mutations in the signaling domains of LapD impacted the interactions between the periplasmic domain of this receptor and LapG.

Consistent with previous experiments, we observed basal levels of cross-linking of LapG to LapD in the absence of c-di-GMP in a UV-dependent manner, however addition of c-di-GMP stimulated complex formation. The basal and stimulated binding is abrogated in the output domain mutants W[125]E and D[72]K, both located at the LapG binding site, corroborating our structural model. These findings also indicate that LapG binding to apo-LapD is still specific, albeit weaker and/or more transient compared to the activated, c-di-GMP-bound state. S helix mutations S[229]D and F[222]E or the HAMP domain mutation I[187]E, which we predict to bias the HAMP domain towards an activated conformation in analogy to orthologous studies on other HAMP domain-containing proteins (*Zhou et al., 2011*), result in a higher basal level of LapG cross-linking, but also an elevated level of cross-linking when c-di-GMP is present (*Figure 9A*).

Another subtle, but mechanistically important difference is apparent when we determined the sensitivity of the mutant alleles to c-di-GMP (*Figure 9B*). Spiking wild-type LapD with increasing amounts of c-di-GMP shows an appreciably sharp transition in LapG binding, with the switch occurring between 0.75 μM and 1 μM dinucleotide. In contrast, the mutant variants of LapD with alterations in the S helix (S[229]D and F[222]E) or the HAMP domain (I[187]E) show a shallower dose–response curve, in addition to the aforementioned higher basal levels (*Figure 9B*). These experiments establish that the deregulation of LapD through destabilization of regulatory interactions in the cytoplasmic domains is a result of altered sensitivity of these mutant c-di-GMP receptors. These studies also reveal a switch-like behavior of the wild-type protein, which suggests cooperativity as an important regulatory feature for LapD activation.

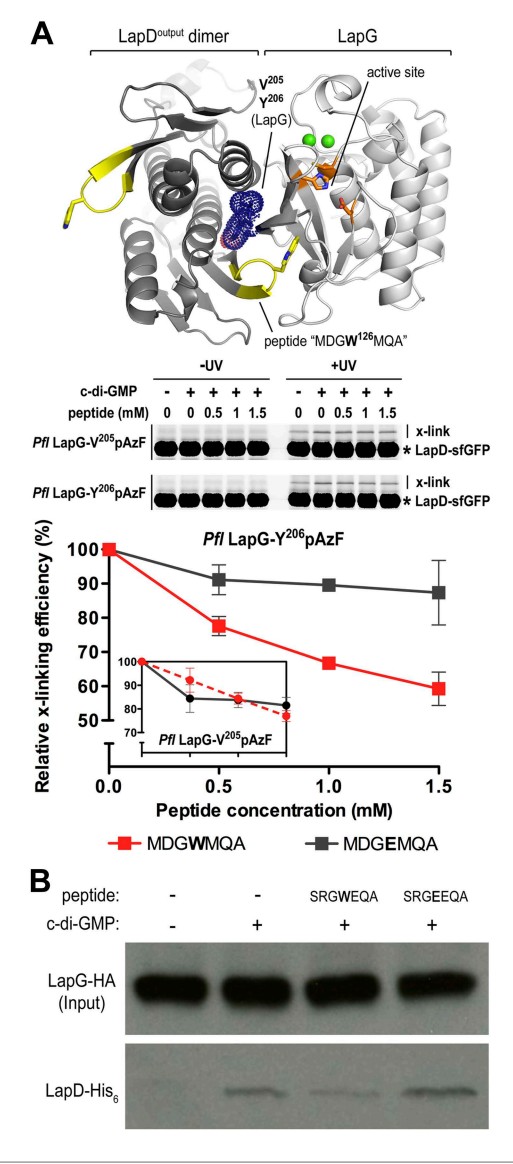

**Figure 8**. Attenuating LapD–LapG interactions with a peptide containing the GWxQ binding motif. (**A**) Cross-linking assay using purified proteins. The CdgS9^Output-LapG crystal structure depicts the GWxQ β-hairpin motif of CdgS9^Output (yellow cartoon) as well as the two sites of pAzF incorporation (blue mesh) on LapG used to assess LapD–LapG interactions (top panel). To assess the effects of the MDGW^126MQA peptide on LapD–LapG interactions, detergent-solubilized, full-length LapD-sfGFP was incubated with purified non-fluorescent LapG containing pAzF at V^205 (inset in bottom panel) or Y^206 (main graph in bottom panel), and increasing concentrations of peptide. Half of the sample was subjected to UV irradiation for 5 min prior to analysis by SDS-PAGE. The average intensity of the fluorescent band corresponding to LapD-sfGFP crosslinked to LapG from three independent measurements was normalized to non-crosslinked LapD-sfGFP, *Figure 8. Continued on next page*

Next, we asked whether LapD regulation in vivo can be altered through mutations in the periplasmic output domain. In particular, we targeted either strictly conserved residues and/or residues that we identified as important for establishing the distinct crystallographic conformations and for the proposed activation mechanism (*Figure 10A*, *Figure 10—figure supplement 1*). In this analysis, we also included a panel of HAMP domain mutations, including I^187E, for which switching phenotypes had been established in chemoreceptors such as TSR (*Figures 10A*, *Figure 10—figure supplement 2*; *Zhou et al., 2011*). As done previously (*Navarro et al., 2011*; *Newell et al., 2011a*), the mutations were introduced in a *lapD/lapG* co-expression plasmid to ensure balanced expression of both genes in *P. fluorescens*. As the read-out, the static biofilm that forms preferentially at the air-liquid interface was quantified using an established crystal violet-based assay. Deletion of *lapD* and *lapG* (Δ*lapD/lapG*) results in a hyperbiofilm phenotype, since the adhesin LapA cannot be processed due to the absence of the protease LapG and thus the adhesin is retained at the cell surface (*Newell et al., 2011a*). On the other end of the spectrum, expression of LapG alone in the double-deletion strain results in hypo-biofilm formation or maximal dispersal, since LapG will process LapA in a constitutive, unregulated fashion. Simultaneous expression of wild-type *P. fluorescens* LapD and LapG in the double-deletion strain restores the phenotype of the parent strain with native LapDG levels, characterized by a low-to-intermediate biofilm mass (*Figure 10A,B*). This range of phenotypes allows us to assess the functionality of mutant constructs. A non-functional LapD variant would phenocopy a double-deletion strain over-expressing only LapG. In contrast, a constitutively active LapD variant would present itself in a hyper-biofilm phenotype, resembling the phenotype of the double-deletion strain. Any phenotype in-between the two extremes would be indicative of a functional LapD protein that retained regulation via c-di-GMP and near-wild-type levels of LapG recruitment.

Considering the group of output domain mutations tested, there is a distinct lack of penetrant phenotypes despite the fact that non-conservative mutations were introduced at pivotal positions based on the structural analysis and sequence alignments (*Figure 10A*, *Figure 10—figure supplement 1*). Some of these mutations would have been expected to be activating (i.e., hyper-biofilm forming) via destabilization of the apo-LapD^Output conformation, but no such phenotypes were

*Figure 8. Continued*

averaged, and plotted as a function of peptide concentration (±SD) (red line; bottom panel). The peptide MDGE$^{126}$MQA was used as a negative control (gray line; bottom panel). (**B**) Pull-downs of LapG-HA from lysates of *P. fluorescens*. LapD-His$_6$ and LapG-HA were co-expressed in a *P. fluorescens* strain with deleted *lapD* and *lapG* genes. Western blots using HA (top panel) and His$_6$ (bottom panel) specific primary antibodies were used to assess the levels of LapD-His$_6$ that co-purify with LapG-HA in the presence or absence of peptides and c-di-GMP.

observed. The more apparent hypo-biofilm phenotypes of mutant LapD variants D$^{72}$K and Y$^{76}$R can be explained by disturbance of the LapG binding interface by these mutations. The weaker phenotypes of the output domain mutants are in stark contrast to the effect of sequence alterations in the HAMP domain. More than half of the single-point mutations made in the HAMP domain result in hyper-biofilm phenotypes indicative of a constitutive LapD–LapG interaction. Of note, the positions in the helical segments of the HAMP domain, which are important for the differential packing interactions within the dimeric HAMP domain (positions *a* and *d* in the knobs-into-holes packing;

*Figure 10*, *Figure 10—figure supplement 2*), have the most pronounced effect on LapD function. These positions are the key elements in the gear-box model for HAMP domain switching (*Hulko et al., 2006*; *Ferris et al., 2011*, *2012*), which describes two states of the HAMP four-helix bundle that are

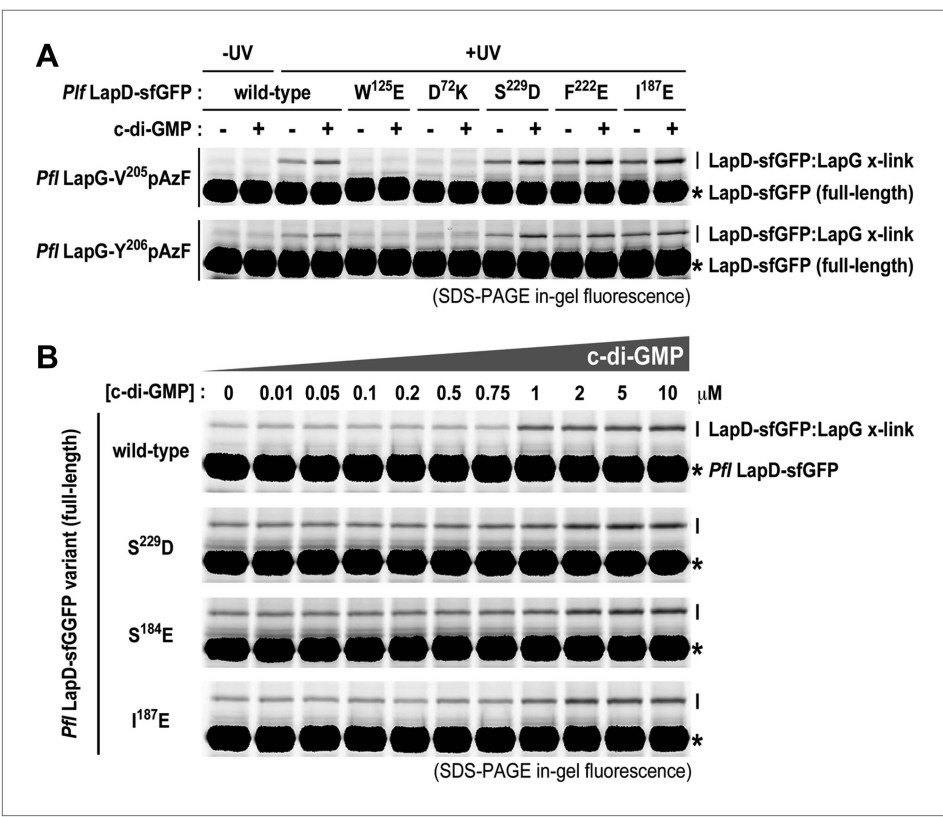

**Figure 9**. Mutations in the HAMP domain and S helix sensitize LapD toward c-di-GMP. (**A**) Effect of structure-guided point mutations on LapD function. Detergent solubilized LapD-sfGFP variants (2 μM) were incubated with non-fluorescent LapG (1 μM) containing pAzF at positions V$^{205}$ (upper gel) and Y$^{206}$ (lower gel) in the presence and absence of c-di-GMP prior to exposure to UV light and SDS-PAGE analysis. Fluorescently imaged gels indicated LapD mutations W$^{125}$E and D$^{72}$K abolish LapD–LapG interactions while HAMP domain and S helix mutations S$^{229}$D, F$^{222}$E, and I$^{187}$E have higher levels of interaction with LapG in the absence and presence of c-di-GMP compared to wild-type. (**B**) C-di-GMP sensitivity of LapD. C-di-GMP was titrated into a fixed amount of detergent solubilized LapD-sfGFP (2 μM) variants and non-fluorescent LapG (1 μM) containing pAzF at position V$^{205}$, after which crosslinking was initiated by UV irradiation and analyzed by SDS-PAGE. A sharp transition in crosslinking is observed between 0.75 μM and 1 μM c-di-GMP for wild-type LapD-sfGFP but not for the other activating mutations, which have higher basal levels of crosslinking and a less pronounced transition.

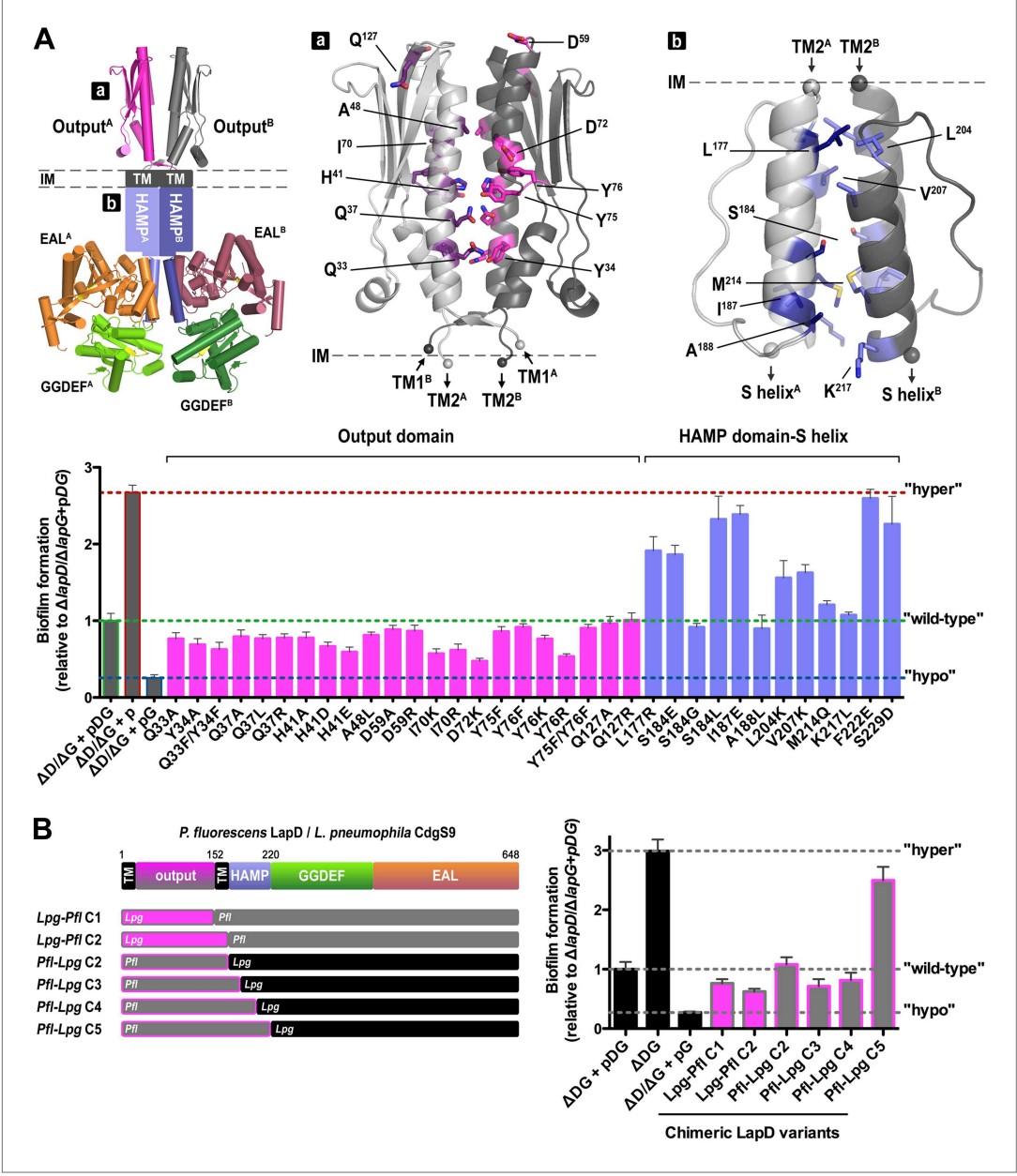

**Figure 10**. Mutational analysis of LapD regulation. (**A**) Biofilm phenotypes of LapD point mutants. A structural model of full-length LapD is shown, which includes the closed, parallel output domain that is linked to the cytoplasmic S helix-GGDEF-EAL domain module via transmembrane and HAMP domains (left panel). Homology models of output domain (middle panel) and a HAMP domain (right panel) of *P. fluorescens* LapD were used to design mutations predicted to alter the activity of LapD. Biofilm phenotypes of these LapD variants were assessed using previously established in vivo assays (lower panel). Representative data from 8–12 wells for at least three biological replicates are shown as means ±SD. (**B**) Biofilm phenotypes of chimeric constructs. LapD constructs were created that contained either the periplasmic or cytoplasmic domains of *L. pneumophila* CdgS9 replacing the respective native *P. fluorescens* LapD domains (left panel). ΔD/ΔG: *P. fluorescens* strain lacking *lapD* and *lapG* genes, pDG: dual expression plasmid for LapD and LapG, p: empty plasmid, pG: expression plasmid for LapG.

The following figure supplements are available for figure 10:

**Figure supplement 1**. Pairwise alignment of LapD and CdgS9.

**Figure supplement 2**. Multiple-sequence alignment of HAMP domains.

separated by 26° rotations of the helices. This observation indicates that LapD's HAMP domain adopts a canonical fold and is likely subject to a similar mode of regulation as has been described for other HAMP domains. Together, these results confirm the regulatory role of the HAMP domain, but also indicate a clear hierarchy of LapD's control elements, with the HAMP domain driving the main regulatory effects, which may be important to bias the system to inside-out signaling and to prevent accidental activation through basal LapD–LapG encounters.

## Conservation of LapD regulation across bacterial species

Conservation of key residues and surfaces does not imply functional conservation; however, our previous results indicated that complexes between the output domain and LapG from two different bacterial strains form readily and in a physiologically relevant manner (*Chatterjee et al., 2012*). To take this point further, and to demonstrate conservation of mechanism and regulation in cells, we created chimeric full-length LapD constructs, in which we replaced the *P. fluorescens* output domain or the cytoplasmic domain with the corresponding domains of CdgS9, the putative *L. pneumophila* LapD ortholog (*Figure 10B*). Constructs *Lpg-Pfl* C1 and C2 contain the *L. pneumophila* output domain with a cytoplasmic module from *P. fluorescens* LapD. *Pfl-Lpg* C2 shares the swap point with *Lpg-Pfl* C2, but is the reciprocal construct consisting of output domain and transmembrane helices of LapD and the entire cytoplasmic fragment of CdgS9. In constructs *Pfl-Lpg* C3, C4, and C5, the chimeric swap was introduced at different positions in the HAMP domain (in-between secondary structure elements). The swap-point in C5 is at the HAMP-S helix junction, which would leave the intra-molecular S helix-GGDEF-EAL control module intact (*Navarro et al., 2011*). The constructs were introduced in the dual expression plasmid that also carried the *P. fluorescens lapG* gene for the balanced expression of both proteins, similar to the native operon. Biofilm phenotypes were assessed as before (see above; *Figure 10A*).

Notably, the majority of our chimeric constructs supported biofilm regulation similar to the levels observed with the native *P. fluorescens* proteins (*Figure 10B*). Importantly, both the output domain and the cytoplasmic HAMP-GGDEF-EAL module from *L. pneumophila* appeared fully functional in this context. The only construct that generated a hyper-biofilm phenotype indicative of a constitutive interaction between LapG and LapD was the one in which the *L. pneumophila* S helix-GGDEF-EAL fragment replaced the respective *P. fluorescens* LapD domains with the HAMP domain coming from *P. fluorescens* (construct *Pfl-Lpg* C5). This result hints at the importance of the connection between the HAMP domain and the S helix. Most importantly for this study, these data demonstrate conservation of the key mechanism between moderately conserved LapD orthologs and indicate that, despite the mixed origin of the complex structure components, CdgS9 is a suitable proxy for a c-di-GMP receptor that mediates inside-out signaling to control periplasmic proteolysis. Consequently, the two crystal structures of the output domain of a LapD-like protein in two distinct conformations, one representing the inactive state, the other the active, LapG-bound state, provide fundamental insights into the switching mechanism.

## Discussion

Biofilm formation in *P. fluorescens* as a response to nutritional cues relies on the *lap* operon, which contains genes encoding the signal-integrating transmembrane protein LapD, a c-di-GMP receptor that regulates periplasmic proteolysis through a sequestration of the protease LapG. LapG engagement is regulated through conformational changes in LapD that originate from c-di-GMP binding to LapD's cytoplasmic domain and are transmitted through the juxtamembrane HAMP domain. Our structural analyses establish that LapD's output domain adopts a PAS domain fold and forms homodimers with distinct differences in the LapG binding-competent, active state and its inactive state (*Video 1*). Notably, the packing transition that the structures depict can be described as a transition from a four-helix to a two-helix bundle, and follows two principle components: in a piston motion, one half-side of the dimeric output domain shifts vertically along the second half-side, whereas scissor-like motions encompass a tilting of the half-sides along the dimer interface. Considering the current model for HAMP domain-switching (*Hulko et al., 2006*; *Airola et al., 2013*), which involves undergoing small amplitude translations and/or rotations during signal transduction similar to the ones we describe for CdgS9's periplasmic output domain, we predict that the output domain follows the conformational changes in the HAMP domain as a direct response. The cross-over of the membrane proximal elements observed in both output domain conformations (*Figure 4B*) may function as the fulcrum about which the output and HAMP domain pivot and/or tilt.

As for the mode of LapG interaction, the four-helix bundle of the output domain of inactive LapD presents a surface that appears suboptimal for LapG binding, while the activated LapD conformation is characterized by extensive shape complementarity and effective interaction anchors at one face of the LapD output domain dimer, foremost the conserved tryptophan in the GWxQ loop. The notion of geometric compatibility, as opposed to an increase in specific per-residue interactions as a predominant driver for high-affinity LapG binding, is supported by the overall low surface conservation of LapD orthologs. The asymmetry of the LapD–LapG complex further suggests mechanisms inherent to the output domain that counteract an activating signal and may ultimately contribute to anti-cooperativity, which has been described for other PAS domain-containing proteins (*Moore and Hendrickson, 2012*). In the current model, (transient) occupation of the second LapG binding site on the dimeric output domain of LapD would release LapG at the first, high-affinity site.

While the crystal structures provide distinct snapshots of possible conformations, they contain little information regarding the dynamics of the transitions and stability of the functional states. Based on our lack of success to create a constitutively active LapD through structure-guided mutations in the output domain, we offer two explanations for the apparent disconnect between the crystallographic data that are based on the isolated output domain and the functional analysis of the regulated full-length protein. In the first model, the regulatory HAMP and GGDEF-EAL domains would constrain the conformation of the LapD output domain to an extent that mutations are insufficient to destabilize the output domain dimer interface and/or induce the correct high-affinity LapG-binding state. Another, non-mutually exclusive model also consistent with our data is that the apo-state may be described as a dynamic ensemble, preventing efficient LapG binding. It would follow that the crystallized output domain in the absence of LapG depicts only one possible conformation of the ensemble with the mutations only having minimal effect on the activity. At the same time, crucial interaction motifs are exposed even in this state, consistent with a basal level of LapG binding in the absence of c-di-GMP. LapD activation by c-di-GMP on the other hand increases the affinity of LapG with the output domain following conformational changes dictated by the cytoplasmic HAMP domain. It is interesting to note that both principles, switching through dynamic states or between more stable conformations have also been discussed for HAMP domains (*Ferris et al., 2011*, *2012*; *Zhu et al., 2013*; *Ames et al., 2014*; *Mechaly et al., 2014*; *Stewart, 2014*), consistent with the idea that the output domain may mirror the behavior of the regulatory domains. Future experiments investigating the dynamics of full-length LapD, including those in a lipid environment (*Wang et al., 2014*), should help to distinguish between the two models.

The dominance of the HAMP domain over the output domain in regulating LapD may bias the system towards inside-out signaling and against tenacious conformational changes induced by LapD–LapG encounters in the periplasm. At the same time, the observed basal and/or transient interactions of LapG with LapD's output domain at low c-di-GMP levels may translate into low frequency conformational changes in the cytoplasmic domains, which would yield pre-activated receptors that would lock in place only when c-di-GMP levels are high. Such a mechanism would explain how c-di-GMP has access to the EAL domain in the first place, given that the autoinhibited conformation is incompatible with c-di-GMP binding. The mechanism would be best described as 'coincidence detection', requiring both c-di-GMP and LapG being available for LapD engagement and full activation. The HAMP domain would function as an attenuator in the inactive state, yet determine the active conformation upon c-di-GMP binding, which would prevent an immediate reversion to the autoinhibited state.

The presence of LapD-like signaling nodes was validated in a few other biofilm-forming bacteria. Unfortunately, while we have evidence that they are regulated in a similar fashion, we know very little about their physiological role. Gene deletion of the *L. pneumophila* LapD ortholog locus *cdgS9* has no obvious phenotype, yet overexpression of CdgS9 has an effect on intracellular growth of the pathogen (*Levi et al., 2011*), suggesting functional relevance of this ortholog. Stronger functional similarities to the *P. fluorescens* system are apparent in more closely related bacteria such as *Pseudomonas putida* (*Gjermansen et al., 2005*, *2006*, *2010*). Future studies will address the physiological pathways that LapD and LapG orthologs regulate. Until then, proteins from these other systems remain orphan c-di-GMP receptors and proteases, respectively. Our work lays the foundation and establishes several tools that will be useful to interrogate these signaling nodes in other bacteria.

As part of these efforts, we demonstrate that, in principle, the LapD–LapG interaction, and likely analogous binding between similar protein pairs, is therapeutically accessible, since LapD-derived

peptides containing the conserved GWxQ motif specifically diminish LapG binding to LapD. While effective competition relied on high peptide concentrations, one needs to consider that we used a non-constrained peptide, which competes with the native binding site presented by full-length, c-di-GMP-activated LapD that presents an interaction surface far greater than just the peptide sequence. Unfortunately, the peptides are not bioavailable and likely do not cross the outer membrane of *P. fluorescens*, preventing us to assess whether inhibiting the LapD–LapG interaction would result in increased biofilm dispersal, as would be predicted if the peptides would release LapG from LapD in vivo. Nevertheless, our data indicate that targeting the interaction site with small-molecule inhibitors could be a fruitful strategy to perturb bacterial biofilm formation and dispersal. Optimized competitors may include synthetic stapled or otherwise designed peptides that mimic the loop structure in solution (*Razavi et al., 2014*).

Based on the conservation of binding events, regulation and key sequence motifs, we predict that our model is universal for proteins with LapD-like architecture, which includes receptors that signal in the opposite direction ('outside-in') by translating environmental signals emerging in the periplasm into changes in DGC and/or PDE activity in their cytoplasmic GGDEF and EAL domains, respectively. As an example, a LapDG-like mode of regulation, yet in the opposite signaling direction and involving an active DGC, controls the clinically important GGDEF domain-containing protein YfiN from *Pseudomonas aeruginosa* and *E. coli* (*Malone et al., 2010*; *Sanchez-Torres et al., 2011*; *Malone et al., 2012*; *Giardina et al., 2013*). In this case, the periplasmic protein YfiR suppresses YfiN's DGC activity through a direct interaction with YfiN's periplasmic PAS domain (*Malone et al., 2012*). YfiR becomes sequestered by the outer membrane protein YfiB, which activates YfiN, likely involving conformational changes in its HAMP-GGDEF domain module (*Malone et al., 2012*; *Raterman et al., 2013*). Recently, a similar outside-in signaling model describing a rheostat-like control mechanism has been established for an autoinducing peptide-sensitive histidine kinase reconstituted into membrane nano-discs (*Wang et al., 2014*). Future studies will ascertain the generality of our model and the potential variations thereof.

## Material and methods

### Protein expression and purification

The periplasmic output domain of *L. pneumophila cdgS9* (*lpg0829*; residues 22–152) and *P. fluorescens lapG* lacking the signal peptide (*Pfl01_0130*; residues 50–251) was amplified from genomic DNA by standard PCR and cloned either separately into a pET28a-based vector (Merck Millipore, Darmstadt, Germany) that adds an N-terminal, cleavable His$_6$-SUMO tag, or together into a bacterial dual expression vector, pETDuet-1 (Merck Millipore, Darmstadt, Germany). In the latter case, *L. pneumophila* His$_6$-SUMO-CdgS9$^{Output}$ and *P. fluorescens* LapG were cloned into multiple cloning site 1 and 2, respectively. *P. fluorescens lapD* (*Pfl01_0131*; residues 1–648) was amplified from genomic DNA by standard PCR and ligated into the *NdeI* and *NotI* restriction sites of a pET24a vector (Merck Millipore, Darmstadt, Germany). To create LapD-sfGFP and LapG-sfGFP fusion proteins, *lapD* was amplified from the pET24a-*lapD* construct and ligated into the *NcoI* and *XhoI* sites of the vector pBrew, which is a pET28a-based vector possessing a 3'-linked tobacco etch virus (TEV) protease cleavage site followed by a superfolder GFP-His$_6$ (sfGFP-His$_6$) gene. Site-directed mutagenesis was carried out using the Quikchange kit (Agilent Technologies, Santa Clara, CA) following the manufacturer's instructions, followed by validation through DNA sequencing. Chimeric LapD constructs were made by overlap-extension PCR.

Native and selenomethionine-substituted proteins were over-expressed in *E. coli* BL21 T7 Express or T7 Express Crystal cells (New England Biolabs, Ipswich, MA), respectively. For the expression of native proteins, cultures were grown at 37°C in Terrific Broth (TB) medium supplemented with 50 µg/ml kanamycin. At an OD$_{600}$ of ~1, the temperature was reduced to 18°C, and protein expression was induced by adding 0.5 mM IPTG. Selenomethionine-labeled proteins were expressed in cells grown at 37°C in M9 minimal medium supplemented with the desired antibiotic, vitamins (1 µg/ml thiamin and 1 µg/ml biotin), carbon source (0.4% glucose), trace elements, and amino acids (50 µg/ml of each of the 20 amino acids with selenomethionine substituting for methionine). Protein expression was induced at an OD$_{600}$ corresponding to ~0.6.

To site-specifically incorporate the cross-linking amino acid *para*-azidophenylalanine (pAzF) into LapD or LapG, an amber (TAG) stop codon was introduced at the desired location within the genes.

BL21 T7 express cells (for pAzF-derivatized LapD expression) or BL21ai cells (Life Technologies, Grand Island, NY; for pAzF-derivatized LapG expression) containing both the amber-disrupted plasmid and the pDule2-pCNF machinery plasmid (*Miyake-Stoner et al., 2009*) were grown in TB medium supplemented or auto-induction medium (*Peeler and Mehl, 2012*), respectively, with 50 μg/ml kanamycin and 100 μg/ml spectinomycin to an $OD_{600}$ of ~0.8, at which time the temperature was dropped to 22°C, and 0.5 mM IPTG and 1 mM pAzF were added.

In all cases, protein expression was allowed to proceed for 16 hr at 18°C, after which cells were harvested by centrifugation, re-suspended in Ni-NTA buffer A (25 mM Tris–HCl [pH 8.5], 500 mM NaCl and 20 mM imidazole; used for LapG, $LapD^{Output}$ and $CdgS9^{Output}$), Ni-NTA buffer A′ (25 mM Tris–HCl [pH 8.5], 200 mM NaCl, and 20 mM imidazole used for the $LapG–CdgS9^{Output}$ complex), or Ni-NTA buffer A″ (25 mM Tris [pH 7.6], 500 mM NaCl, and 10% glycerol; used for full-length LapD), and flash frozen in liquid nitrogen.

For soluble proteins, cell suspensions were thawed and lysed by sonication. Cell debris was removed by centrifugation and the clear lysates were incubated with NiNTA resin (Qiagen, Valencia, CA) that was pre-equilibrated with Ni-NTA buffer A. The resin was washed with 20 column volumes of Ni-NTA buffer A, followed by protein elution with five column volumes of Ni-NTA buffer B (25 mM Tris–HCl [pH 8.5], 500 mM NaCl, and 300 mM imidazole). The eluted proteins were buffer-exchanged into gel filtration buffer (25 mM Tris–HCl [pH 8.5] and 150 mM NaCl) on a fast desalting column (GE Healthcare, Little Chalfont, UK). Proteins were subjected to size exclusion chromatography on a Superdex 200 column (GE Healthcare, Little Chalfont, UK) pre-equilibrated with gel filtration buffer. Where indicated, the $His_6$-SUMO moiety was cleaved off by using the yeast protease Ulp-1 following the desalting step. Ulp-1, uncleaved protein and the cleaved fusion tags were removed by NiNTA affinity chromatography prior to the final gel filtration. For co-purification of the $CdgS9^{Output}$–LapG complex, the procedure remained identical except that all buffers contained 200 mM NaCl instead of the amount indicated above. Purified proteins were concentrated on Amicon filters with an appropriate size cutoff to concentrations >30 mg/ml, flash frozen in liquid nitrogen and stored at −80°C.

To isolate membranes containing LapD (or LapD-sfGFP), 0.5 mg/ml lysozyme was added to thawed cells, which were then lysed by sonication. Unlysed cells were pelleted by centrifugation at 5000×*g* for 5 min. Membranes were pelleted by centrifuging the supernatant at 180,000×*g* for 1 hr. The resulting supernatant was decanted and the membrane pellet was resuspended in membrane buffer (25 mM Tris [pH 7.5], 500 mM NaCl, and 10% glycerol), sonicated briefly to homogenize the membranes and frozen in liquid $N_2$. Where specified, LapD was solubilized for subsequent purifications by incubating membranes in membrane buffer supplemented with 2% Triton X-100 (Sigma-Aldrich, St. Louis, MO). After 1 hr, insoluble material was pelleted by centrifugation at 180,000×*g* for 40 min. The clarified supernatant was incubated with NiNTA resin for 1 hr and washed thoroughly with membrane buffer supplemented with 4 mM β-mercaptoethanol (BME), 20 mM imidazole, and 1% Triton X-100. The Triton X-100 detergent was exchanged for 0.01% (wt/vol) lauryl maltose neopentyl glycol (LMNG) in a step-wise fashion while LapD was still bound to the NiNTA resin, followed by elution of LapD with LapD elution buffer (25 mM Tris [pH 7.6], 500 mM NaCl, 10% glycerol, 4 mM BME, 300 mM imidazole, and 0.01% LMNG). The eluted protein was buffer-exchanged into a low-salt LapD buffer (25 mM Tris–HCl [pH 7.5], 250 mM NaCl, 5% glycerol, 4 mM BME, and 0.01% LMNG) on a fast desalting column. LapD was concentrated by ultra-filtration and subjected to size exclusion chromatography on a Superdex 200 column pre-equilibrated with LapD gel filtration buffer (25 mM Tris–HCl [pH 7.6], 150 mM NaCl, 5% glycerol, 4 mM BME, and 0.0025% LMNG and 0.02% CHAPS). The protein was stored at 4°C and used within 72 hr.

To prepare LapD-sfGFP for cross-linking assays, isolated membranes were solubilized in membrane buffer supplemented with 1% lauryl maltose neopentyl glycol (LMNG) and 0.2% CHAPS for 1 hr at 4°C, followed by centrifugation at 100,000×*g* for 30 min. The concentration of solubilized LapD-sfGFP (and LapG-sfGFP) was determined by comparing the amount of fluorescence in the supernatant to a standard curve of purified sfGFP (excitation: 488 nm, emission: 510 nm). Proteins were used in cross-linking assays immediately after solubilization.

To produce non-fluorescent LapG with pAzF, purified LapG-sfGFP was incubated with TEV protease overnight at 4°C. The $His_6$-tagged sfGFP and TEV protease were subsequently removed by incubation with NiNTA resin. Unbound, non-fluorescent LapG containing site-specifically incorporated pAzF was collected in the flow through, concentrated to >2 mg/ml, frozen in liquid nitrogen, and stored at −80°C.

## Crystallization, data collection and structure solution

Crystals were obtained by hanging drop vapor diffusion mixing equal volumes of protein (10–30 mg/ml) and reservoir solution followed by incubation at 20°C. For the native and selenemethionine-derivatized *L. pneumophila* CdgS9$^{Output}$ crystals, the reservoir solution contained 0.1 M Bis-Tris (pH = 5.0), 14% PEG3350, and 4% vol/vol 2,2,2-trifluoroethanol. Crystals of the native LapG–CdgS9$^{Output}$ complex were obtained from a reservoir solution containing 0.1 M Bis-Tris (pH = 6.0) and 0.1 M magnesium formate. Our initial efforts directed towards experimental phase determination proved intractable due to extremely low expression levels of the selenomethione-substituted LapG–CdgS9$^{Output}$ complex. Hence phases were determined based on data sets collected on crystals grown from protein mixtures containing selenomethionine-derivatized *P. fluorescens* LapG and native *L. pneumophila* CdgS9$^{Output}$, which grew at conditions identical to the native complex. For cryo-protection, crystals were soaked in reservoir solution supplemented with 20–25% xylitol. Cryo-preserved crystals were flash-frozen and stored in liquid nitrogen. Data were collected on frozen crystals at 100 K using synchrotron radiation at the Cornell High Energy Synchrotron Source (CHESS, Ithaca).

Data reduction was carried out with the software package HKL2000 (*Otwinowski and Minor, 1997*). Experimental phases for the initial structure determination were obtained from Single Anomalous Diffraction (SAD) experiments on crystals grown from selenomethionine-derivatized proteins by using the software package PHENIX (*Adams et al., 2002*). Refinement in PHENIX and COOT (*Emsley and Cowtan, 2004*) yielded the final models. Data collection and refinement statistics are summarized in *Table 1*. Illustrations were made in Pymol (Schrödinger). Alignments were generated using ClustalW2 (*Larkin et al., 2007*). Homology models were created using the program Modeller (*Eswar et al., 2006*) using default inputs. Crystallographic software is provided through SBGrid (*Morin et al., 2013*).

## Crosslinking of LapD and LapG

Unless specified otherwise, LapG–LapD cross-linking was performed with 1 µM purified LapG (either fused to sfGFP or not) containing pAzF at the desired locations and solubilized variants of LapD at 2 µM final concentration (either fused to sfGFP or not) in binding buffer (25 mM Tris [pH 7.5], 250 mM NaCl, 2.5 mM CaCl$_2$, 0.01% LMNG, and 0.2% CHAPS). When indicated, 10 µM c-di-GMP was added to the reaction. All reaction components were allowed to incubate at room temperature for 20 min, at which time crosslinking was initiated by irradiation with short wave UV light (~254 nm) for 5 min. After irradiation, samples were mixed with SDS sample buffer and immediately subjected to SDS-PAGE without boiling. Gels were imaged by fluorescence using a BioRad Gel Doc system. Band intensities of the cross-linked adducts were measured using ImageLab software (Biorad, Hercules, CA) and normalized to the intensity of non-cross-linked sfGFP-labeled protein. Reported values of band intensities are the average of three replicates.

## Size exclusion chromatography-coupled multi-angle light scattering (SEC-MALS)

SEC-MALS measurements (*De et al., 2010*) were carried out by injecting purified proteins (100 µM) onto a Phenomenex gel filtration column pre-equilibrated with MALS buffer (25 mM Tris–HCl [pH 7.4] and 100 mM NaCl). The chromatography system was coupled to an 18-angle, static light scattering detector and a refractive index detector (DAWN HELEOS-II and Optilab T-rEX, respectively; Wyatt Technology, Santa Barbara, CA). Data were collected at 25°C every second at a flow rate of 1 ml/min and analyzed with the software ASTRA, yielding the molecular weight and mass distribution (polydispersity) of the samples. For data quality control and normalization of the light scattering detectors, monomeric bovine serum albumin (Sigma-Aldrich, St. Louis, MO) was used.

## Growth conditions

*P. fluorescens* was routinely cultured with liquid LB medium or on solidified LB with 1.5% agar at 30°C. Gentamicin (Gm) was used at 30 µg/ml. To induce expression of the P$_{BAD}$ promoter from pMQ72, arabinose was added at 0.2% (vol/vol). For biofilm and pull down assays, *P. fluorescens* was grown in K10T-1 medium, previously described as a phosphate-rich medium (*Monds et al., 2006*). K10T-1 consists of 50 mM Tris–HCl (pH 7.4), 0.2% (wt/vol) Bacto tryptone, 0.15% (vol/vol) glycerol, 0.61 mM MgSO$_4$, and 1 mM K$_2$HPO$_4$.

## Biofilm assay

Static biofilm formation assays of *P. fluorescens* were performed as previously described (*Newell et al., 2011b*). Polyvinyl chloride 96-well round bottom microtiter plates (Corning, Corning, NY), containing 100 µl of K10T-1/well were inoculated with 1.5 µl of an overnight LB liquid culture of *P. fluorescens*. Biofilm assays were statically grown at 30°C. After 6 hr of growth, unattached bacteria and medium were removed and the wells were stained with 0.1% (wt/vol) solution of crystal violet in water. To quantify biofilm biomass, crystal violet was solubilized in a solution containing 30% methanol and 10% acetic acid and a Spectra Max M2 microplate reader (Molecular Devices, Sunnyvale, CA) was used to read the absorbance at 550 nm, as previously described (*Monds et al., 2007*). Quantification of 8–10 wells for at least three biological replicates was obtained for each *P. fluorescens* strain assessed.

## Pull-down assay from *P. fluorescens* lysates

Peptide competition for the native LapD–LapG interaction was assessed using an established pull-down assay (*Newell et al., 2011a*). *P. fluorescens*-clarified cell extracts co-expressing LapD-His$_6$ and LapG-HA were prepared in pull down buffer consisting of 20 mM Tris (pH 8.0), 10 mM MgCl$_2$ and in the presence of 10 µM chemically pure c-di-GMP (GLSynthesis Inc., Worcester, MA) and 1 mM custom manufactured peptides (Elim Biopharm). Immunoprecipitations contained 450 µl of clarified cell extract, 40 µl Protein A sepharose (Genscript, Piscataway, NJ), 0.8% thesit, and 0.5 µl monoclonal, mouse anti-HA antibody (Covance, Princeton, NJ). Immunoprecipitations were incubated at 4°C for 60 min and then washed three times at room temperature with gentle shaking in pull down buffer with the addition of c-di-GMP to a final concentration of 10 µM. Immunoprecipations were analyzed by SDS-PAGE and Western blotting to detect LapD-His$_6$.

## Acknowledgements

We thank Michael Airola for fruitful discussions. Part of this work is based upon research conducted at the Cornell High Energy Synchrotron Source (CHESS), which is supported by the National Science Foundation and the National Institutes of Health/National Institute of General Medical Sciences under NSF award DMR-1332208, using the Macromolecular Diffraction at CHESS (MacCHESS) facility, which is supported by award GM-103485 from the National Institute of General Medical Sciences, National Institutes of Health. Our work was supported by the NIH under grants R01-AI097307 (HS/GAO), F32-GM108440 (RBC), and T32-GM08704 (CDB), by the NSF under grant MCB9984521 (GAO), and by a PEW scholar award in Biomedical Sciences (HS).

## Additional information

### Funding

| Funder | Grant reference number | Author |
| --- | --- | --- |
| National Science Foundation | DMR-1332208 (CHESS) | Holger Sondermann |
| National Institutes of Health | GM-103485 (MacCHESS) | Holger Sondermann |
| National Institutes of Health | R01-AI097307 | George A O'Toole, Holger Sondermann |
| National Institutes of Health | F32-GM108440 | Richard B Cooley |
| National Institutes of Health | T32-GM08704 | Chelsea D Boyd |
| National Science Foundation | MCB9984521 | George A O'Toole |
| Pew Charitable Trusts | | Holger Sondermann |

The funders had no role in study design, data collection and interpretation, or the decision to submit the work for publication.

### Author contributions

DC, RBC, CDB, Conception and design, Acquisition of data, Analysis and interpretation of data, Drafting or revising the article; RAM, Drafting or revising the article, Contributed unpublished essential data or reagents; GAO'T, HS, Conception and design, Analysis and interpretation of data, Drafting or revising the article

# Additional files

## Major datasets

The following datasets were generated:

| Author(s) | Year | Dataset title | Dataset ID and/or URL | Database, license, and accessibility information |
|---|---|---|---|---|
| Chatterjee D, Cooley RB, Boyd CD, Mehl RA, O'Toole GA, Sondermann HS | 2014 | Structure of the periplasmic output domain of the Legionella pneumophila LapD ortholog CdgS9 in the apo state | http://www.rcsb.org/pdb/explore/explore.do?structureId=4u64 | Publicly available at RCSB Protein Data Bank. |
| Chatterjee D, Cooley RB, Boyd CD, Mehl RA, O'Toole GA, Sondermann HS | 2014 | Structure of the periplasmic output domain of the Legionella pneumophila LapD ortholog CdgS9 in complex with Pseudomonas fluorescens LapG | http://www.rcsb.org/pdb/explore/explore.do?structureId=4u65 | Publicly available at RCSB Protein Data Bank. |

The following previously published datasets were used:

| Author(s) | Year | Dataset title | Dataset ID and/or URL | Database, license, and accessibility information |
|---|---|---|---|---|
| Navarro MV, Newell PD, Krasteva PV, Chatterjee D, Madden DR, O'Toole GA, Sondermann H | 2011 | Structure of Pseudomonas fluorescence LapD periplasmic domain | http://www.rcsb.org/pdb/explore/explore.do?structureId=3pjv | Publicly available at RCSB Protein Data Bank. |
| Chatterjee D, Boyd CD, O'Toole GA, Sondermann H | 2012 | Legionella pneumophila LapG (calcium-bound) | http://www.rcsb.org/pdb/explore/explore.do?structureId=4fgo | Publicly available at RCSB Protein Data Bank. |
| Ferris HU, Dunin-Horkawicz S, Mondejar LG, Hulko M, Hantke K, Martin J, Schultz JE, Zeth K, Lupas AN, Coles M | 2011 | The solution structure of the HAMP domain of the hypothetical transmembrane receptor Af1503 | http://www.rcsb.org/pdb/explore/explore.do?structureId=2l7h | Publicly available at RCSB Protein Data Bank. |
| Airola MV, Watts KJ, Bilwes AM, Crane BR | 2010 | Crystal structure of poly-HAMP domains from the P. aeruginosa soluble receptor Aer2 | http://www.rcsb.org/pdb/explore/explore.do?structureId=3lnr | Publicly available at RCSB Protein Data Bank. |

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
