## [Decision Letter]

Thank you for sending your work entitled “Mechanistic insight into the conserved allosteric regulation of periplasmic proteolysis by c-di-GMP” for consideration at *eLife.* Your article has been favorably evaluated by Michael Marletta (Senior editor), a member of our Board of Reviewing Editors, and 2 reviewers.

The Reviewing editor and the reviewers discussed their comments before we reached this decision, and the Reviewing editor has assembled the following comments to help you prepare a revised submission.

The reviewers were generally enthusiastic about your manuscript and felt that it provided new and important insights into signaling through membrane proteins in bacteria. However they felt that there was a disconnect between the structural analyses that are reported and some of the statements, especially about signal transduction from cytoplasm to periplasm. The decision was to ask for a modified manuscript in which these issues are addressed, either by toning down the language, providing amplified discussions of the caveats and limitations of the structural studies, or providing biochemical evidence. Some of the issues noted in the reviews:

1) Need to provide a fuller discussion of the failure of mutations to destabilize the LapD-output domain and its implications for your model.

2) Need to provide an expanded justification for the relevance of CdgS9 as a replacement for LapD.

3) Address the implications for the missing HAMP domain in your structures by providing a clearer hypothesis based possibly on other published structures about how the structural information inside the cell is communicated to the outside.

4) Consider using biochemical assays using liposomes and nano disk-type systems as these systems have provided valuable information on GPCRs (Wang et al. Cell 2014 53:929-940).

The full reviews are included to provide more detail.

Reviewer #1:

This manuscript describes the structure-function study of the LapD/LapG system, in which the level of c-di-GMP is communicated via LapD (c-di-GMP-binding transmembrane protein) to the periplasm. The authors provide new key mechanistic insights into how these signals are actually communicated structurally from the cytosol to the periplasm and especially the changes in the output domain that allow it to go from a weak LapG-binding to a tight LapG binding state. Furthermore, these structures provide novel approaches for targeting this interaction in order to inhibit biofilm formation, one of which (the use of peptides as mimics of a protein interface), was tested directly by the authors.

In short, the manuscript is very well-written, insightful and compelling. Its only central shortcoming is that it does not yet provide the structure(s) of what the authors clearly demonstrate is the key mediator of the structural signaling across the membrane: the HAMP domain (i.e., “with the HAMP domain driving the main regulatory effects”). A structure of a LapD with the HAMP domain will likely be able to provide these key insights and is eagerly awaited. However, even without this piece of this story, the discoveries described here advance our understanding of this 'inside-out' signaling system and will be of interest to a broad range of readers.

There are some concerns, as described in detail below, which should be addressed prior to publication.

The model put forward by the authors is that there is a conformational change in the LapD output domain that mediates a transition between a LapG low and high affinity state. This is based largely on the conformational changes observed between apo and LapG-bound LapD output domain. While it is highly likely that the complex is a single stable state, one might ask whether or not in the absence of LapG, the output dimer is actually (somewhat) dynamic. SAXS would be one potential way to get at this question. This is asked in part to address the observation that the mutations generated to 'destabilize' the apo-LapD-output domain did not make it constitutively active as predicted. That is, perhaps this conformation crystallized is not the single 'stable' conformation for the apo-state in vivo and thus mutations that destabilize it have only a minimal effect on activity.

The superposition shown in Figure 4 shows the elements anchored to the membrane are in nearly identical positions. This makes it difficult to rationalize how the changes from a 'low-to-high' affinity state are communicated via LapD through the membrane, in some ways the central crux of this model. More detailed discussion about how the changes observed may be communicated through the membrane is warranted.

Calcium binding: has Ca been extracted (EGTA) and then ITC performed to quantify the binding and stoichiometry? This should be rather straightforward and would confirm the two versus one binding site question between the two LapG homologs.

Please calculate the shape complementarity between the LapD dimer and LapG & also for the second LapG in the model in Figure 7.

Reviewer #2:

This manuscript describes a membrane-associated c-di-GMP regulated system in signal transduction. The authors work primarily with the LapD/LapG system from Pseudomonas, and also with the LapD homolog CdgS9 from L pneumophila. The P fluorescens system is involved in biofilm formation, and these LapD/LapG systems are present in many different bacterial species. The important aspect of this work is that they use a lot of structural and biochemical studies to map LapD (or in the case of the structural studies CdgS9) with LapG. LapD is a transmembrane protein that binds c-di-GMP in its cytoplasmic domain to modulate its periplasmic domain binding to the LapG protease. The authors showed that they can exchange LapD by CdgS9. There is a lot of work performed here, and some nice structural and biochemical studies. My main concerns here are:

1) The authors could only co-crystalize CsgS9 from L pneumophila with the LapD from P fluorescences. This makes me a bit nervous on how accurate the predictions on functions of homologous pairs mirrored the non-homologous pair, especially because some of the main differences between LapD and CdgS9 localize to the site of interaction with LapD.

2) The authors start the manuscript on Figure 1 showing two very different crystal structures for LapD and CdgS9, and then do crosslinking experiments to suggest that their previous LapD crystal structure is not in the right conformation, which should be that of CdgS9. That made me nervous.

3) How can you say that c-di-GMP triggers the conformational change in the periplamsic domain, when most studies were performed with just the periplasmic domain of LapD and not full length?

4) The fact that a CdgS9 mutant has no phenotype.

---

## [Author Response]

The reviewers were generally enthusiastic about your manuscript and felt that it provided new and important insights into signaling through membrane proteins in bacteria. However they felt that there was a disconnect between the structural analyses that are reported and some of the statements, especially about signal transduction from cytoplasm to periplasm. The decision was to ask for a modified manuscript in which these issues are addressed, either by toning down the language, providing amplified discussions of the caveats and limitations of the structural studies, or providing biochemical evidence. Some of the issues noted in the reviews:

We have revised the manuscript following the specific feedback provided by the reviewers. We hope that our edits clarify the issues that were brought up during the review.

*1) Need to provide a fuller discussion of the failure of mutations to destabilize the LapD-output domain and its implications for your model*.

We added clarifying statements in the corresponding text. Briefly, the mutants behave similar to the wild-type protein, retaining LapD function, which results in an intermediate level of biofilm formation. Although this implies that the proteins are made and intact (otherwise we would observe hyper-biofilm phenotypes), it thus appears that, based on our experiments with full-length proteins, that the crystallographic model does not fully explain the low-affinity state for LapG. We incorporate this point explicitly into the revised manuscript.

Reviewer 1 suggested a model that is identical to the one we proposed in the Discussion (4^th^ paragraph). It appears that the reviewer had no issues with our interpretation of the data. We discuss a second, non-mutually exclusive model in the revised manuscript. In this context, we acknowledged the limitations of the crystallographic analysis, which was the main motivation for investigating the regulation of the full-length protein. While the design of the functional studies was heavily informed by the structures, the discussion of the signaling mechanisms is predominantly based on studies using the full-length protein or cellular assays. We feel that we were careful not to over-interpret the structural data, but rather incorporate our biochemical insights in our analysis and interpretations. The resulting functional model is a composite of the different approaches: we feel that combining information from structural studies with assays using full-length proteins *in vitro*, plus genetic studies *in vivo*, to build our overall model is one of the strengths of our study.

*2) Need to provide an expanded justification for the relevance of CdgS9 as a replacement for LapD*.

While we were only able to obtain crystals of a heterologous complex (*L. pneumophila* CdgS9 output domain and *P. fluorescens* LapG), we would like to point out that we provide ample evidence at several levels that our structural model reveals conserved and relevant features. Simply stated, it was the heterologous complex that provided the structural data, and we leveraged this information to test hypotheses about the structural model in our well-characterized LapD-LapG system in *P. fluorescens*. We make an explicit statement to address this point in the revised manuscript. Please see our response to Reviewer 2, point #1 for more details.

In addition, we showed in a previous publication that the isolated output domain orthologs bind their corresponding LapG orthologs, but also show significant cross-specificity. We refer to this work in our manuscript. The use of orthologous proteins as surrogates is common practice to obtain structural models, a point we now mention in the revised manuscript. Our validation experiments comparing the heterologous and homologous complexes provide confidence that the structural model reports on a native-like interaction. We also would like to point out that all functional assays were conducted with the homologous system from *P. fluorescens*.

*3) Address the implications for the missing HAMP domain in your structures by providing a clearer hypothesis based possibly on other published structures about how the structural information inside the cell is communicated to the outside*.

We included a figure supplement to Figure 10, showing a sequence alignment between LapD’s HAMP domain and that of HAMP domains with known structures, several of which have been investigated by mutagenesis approaches as well. While LapD’s HAMP domain appears more divergent, key residues involved in the proposed switching mechanism of canonical HAMP domains are conserved, and it is those residues that we targeted for our analysis. We expanded the section regarding a potential model for inside-out signaling via the HAMP domains, as described below (response to Reviewer #1, points #2 and #3). The observation that the output domains cross-over right at the membrane surface suggests a scissor-type mechanism where a rotation and/or pivoting of the HAMP and transmembrane domains is coupled to similar conformational changes within the output domain dimer.

*4) Consider using biochemical assays using liposomes and nano disk-type systems as these systems have provided valuable information on GPCRs (Wang et al. Cell 2014 53:929-940)*.

While nano-discs and proteoliposomes are excellent systems for biophysical approaches that we plan to employ in the future to study the dynamics of LapD in more detail, we demonstrate here (and previous work from our groups) that studies on detergent-solubilized LapD correlate well with our functional assessment of the LapD in cell-based assays. In particular, we observe robust and strictly c-di-GMP-dependent switching between the low- and high-affinity state with regard to LapG binding *in vitro*, a finding that has been corroborated with cell-based studies. Targeted mutations based on structural models of the control elements of LapD override regulatory features, both using *in vitro* and cell-based assays, further attesting to the validity of our approach and results. However, we discuss the potential of membrane-reconstituted systems in the revised manuscript, and refer to the studies on a histidine kinase, which are relevant in the context of our work.

It was suggested that we revise the title to “Mechanistic insight into the conserved allosteric regulation of periplasmic proteolysis by the signaling molecule cyclic-di-GMP”, and we agree that this title helps clarify the central points of the paper.

Reviewer #1:

*[…] In short, the manuscript is very well-written, insightful and compelling. Its only central shortcoming is that it does not yet provide the structure(s) of what the authors clearly demonstrate is the key mediator of the structural signaling across the membrane: the HAMP domain (i.e., “with the HAMP domain driving the main regulatory effects”). A structure of a LapD with the HAMP domain will likely be able to provide these key insights and is eagerly awaited. However, even without this piece of this story, the discoveries described here advance our understanding of this 'inside-out' signaling system and will be of interest to a broad range of readers*.

We agree with this reviewer that it would be wonderful to obtain a structural view of the HAMP domain of LapD. Unfortunately, this goal is not within reach at the moment. Nevertheless, the homology models and structure-guided mutagenesis approach used here suggests that LapD’s HAMP domain is a dominant functional unit for LapD regulation.

*There are some concerns, as described in detail below, which should be addressed prior to publication*.

*The model put forward by the authors is that there is a conformational change in the LapD output domain that mediates a transition between a LapG low and high affinity state. This is based largely on the conformational changes observed between apo and LapG-bound LapD output domain. While it is highly likely that the complex is a single stable state, one might ask whether or not in the absence of LapG, the output dimer is actually (somewhat) dynamic. SAXS would be one potential way to get at this question. This is asked in part to address the observation that the mutations generated to 'destabilize' the apo-LapD-output domain did not make it constitutively active as predicted. That is, perhaps this conformation crystallized is not the single 'stable' conformation for the apo-state in vivo and thus mutations that destabilize it have only a minimal effect on activity*.

The model proposed by this reviewer is described in the 4^th^ paragraph of the Discussion. We did not mention it as part of sections describing the crystallographic results since it is based on the functional and mutational analysis introduced later in the manuscript. We extended this discussion, also in response to point #1 in the editor’s letter.

We have considered experiments to investigate the dynamics of the isolated output domain. However, as discussed in the manuscript, our data indicate that the isolated output domain is unregulated since it readily binds LapG (in contrast to the full-length protein which requires c-di-GMP binding to the EAL domain for high affinity LapG binding). Hence, SAXS data collected on the isolated output domain may not be relevant or informative for the native functioning of the LapD protein. Also, we suspect that the conformational changes or ensemble may be too subtle to be picked up reliably by SAXS. For these reasons, we decided to shift the focus to the full-length protein. Future studies will address the dynamics of LapD, in particular of the output and HAMP domains in the context of the full-length protein, and address the molecular basis for the switching mechanism elucidated in the current manuscript.

*The superposition shown in*
Figure 4
*shows the elements anchored to the membrane are in nearly identical positions. This makes it difficult to rationalize how the changes from a 'low-to-high' affinity state are communicated via LapD through the membrane, in some ways the central crux of this model. More detailed discussion about how the changes observed may be communicated through the membrane is warranted*.

We added the following sentence to the Discussion (1^st^ paragraph), which hopefully clarifies our model in this regard: “The cross-over of the membrane proximal elements observed in both output domain conformations (Figure 4) may function as the fulcrum about which the output and HAMP domain pivot.”

*Calcium binding: has Ca been extracted (EGTA) and then ITC performed to quantify the binding and stoichiometry? This should be rather straightforward and would confirm the two versus one binding site question between the two LapG homologs*.

We have attempted the proposed experiment but unfortunately, EGTA-treatment destabilizes *P. fluorescens* LapG rendering the ITC experiments inaccessible. While this result prevents us from quantifying binding and stoichiometry, it suggests a critical role of calcium ions for LapG’s structural integrity.

*Please calculate the shape complementarity between the LapD dimer and LapG & also for the second LapG in the model in*
Figure 7.

As requested, we report calculated values for shape complementarity in the revised manuscript.

Reviewer #2:

*1) The authors could only co-crystalize CsgS9 from L pneumophila with the LapD from P fluorescences. This makes me a bit nervous on how accurate the predictions on functions of homologous pairs mirrored the non-homologous pair, especially because some of the main differences between LapD and CdgS9 localize to the site of interaction with LapD*.

We shared the initial concern with reviewer 2, which is the reason why we validated the structural models and demonstrated conserved functionality in great detail and at several levels. Figures 6 and 7 show complex formation via similar surfaces and with similar stoichiometries. Figure 10 shows functionality of the domain in a cell-based assay and a model system for biofilm formation. While we may miss nuances of the interaction by not having a structural model of the homologous complex, the core machinery relevant for function appears to be well conserved. The homologous complex may resist crystallization for several reasons, which often remain elusive. In this regard, it is important to point out that the complexes (homologous and heterologous) form *in vitro*, and are dependent on the strictly conserved tryptophan residue (see [9]). As noted in the manuscript, we also identify based on the heterologous complex structure a second residue on LapD (D^72^K; Figure 10) that is crucial for the LapD-LapG interaction in *P. fluorescens*, further validating the structural model.

*2) The authors start the manuscript on*
Figure 1
*showing two very different crystal structures for LapD and CdgS9, and then do crosslinking experiments to suggest that their previous LapD crystal structure is not in the right conformation, which should be that of CdgS9. That made me nervous*.

Our data presented here strongly suggests that the *P. fluorescens* output domain crystallized in an aberrant state. We came to this conclusion based on the observation that the two independent CdgS9 output domain-containing structures (apo and LapG-bound) revealed a similar topology compared to our previously determined crystal structure of *P. fluorescens* LapG’s output domain. The CdgS9 topology is also consistent with the canonical PAS fold. More importantly, our validation by site-directed cross-linking leaves little doubt that the output domain in full-length LapD adopts a similar conformation.

However, it is worth noting that the model we published previously is correct in that the electron density and resulting structure were interpreted correctly. We considered the possibility that the two orthologs fold differently, or that they represent different activity states of the output domain, but our biochemical assessment indicated that the LapD output domain must have crystallized in some off-pathway state (Figure 2). Together, the evidence strongly supports that nature (or at least our expression/crystallization conditions) played a trick on us with the previously published isolated output domain of LapD.

3) How can you say that c-di-GMP triggers the conformational change in the periplamsic domain, when most studies were performed with just the periplasmic domain of LapD and not full length?

The sentence prior to this section refers to the overall model we propose for LapD regulation based on our (previous and current) data, which we then describe in the section the reviewer refers to. While studies described up to this point of the manuscript focused on the isolated periplasmic domain, the majority of experiments in the second half of the manuscript were performed with the full-length protein and support the aforementioned model for c-di-GMP regulation (e.g. Figure 7).

*4) The fact that a CdgS9 mutant has no phenotype*.

We agree that it would be desirable to identify the functional role of CdgS9 in *L.* the *pneumophila*. Currently, we treat this ortholog as a vehicle to demonstrate conservation of overall mechanism and as a surrogate for structural studies. We would like to note that *L. pneumophila* has a complex life style, and it is not clear at the moment when this signaling system is expressed. However, it is clear that the regulatory and output domains of CdgS9 can replace the corresponding *P. fluorescens* LapD domains in our chimeric proteins, strongly suggesting that they retain similar functions (Figure 10).